



# Landsat, MODIS, and VIIRS snow cover mapping algorithm performance as validated by airborne lidar datasets

Timbo Stillinger[1], Karl Rittger[1,2], Mark S. Raleigh[3], Alex Michell[1], Robert E. Davis[4], and Edward H. Bair[1]

[1]Earth Research Institute, University of California at Santa Barbara, Santa Barbara, CA 93106, USA
[2]Institute of Arctic and Alpine Research, University of Colorado at Boulder, Boulder, CO 80309, USA
[3]College of Earth, Ocean, and Atmospheric Sciences, Oregon State University, Corvallis, OR 97331, USA
[4]Cold Regions Research and Engineering Laboratory, Hanover, NH 03755, USA

*Correspondence to*: Timbo Stillinger (tcs@ucsb.edu)

**Abstract.** Snow cover mapping algorithms utilizing multispectral satellite data at various spatial resolutions are available, each treating subpixel variation differently. Past evaluations of snow mapping accuracy typically relied on satellite data collected at a higher spatial resolution than the data in question. However, these optical data cannot characterize snow cover mapping performance under forest canopies or at the meter scale. Here, we use 3 m spatial resolution snow depth maps collected on 116 days by an aerial laser scanner to validate band ratio and spectral mixture

snow cover mapping algorithms. Such a comprehensive evaluation of sub-canopy snow mapping performance has not been undertaken previously. The following standard (produced operationally by an agency) products are evaluated: NASA gap-filled Moderate-resolution Imaging Spectroradiometer (MODIS) MOD10A1F, NASA gap-filled Visible Infrared Imaging Radiometer Suite (VIIRS) VNP10A1F, and USGS Landsat 8 Level-3 Fractional Snow Covered Area. Two spectral unmixing approaches are also evaluated: Snow Covered Area and Grain size (SCAG) and Snow

Property Inversion from Remote Sensing (SPIReS), both of which are gap-filled MODIS products and are also run on Landsat 8. We assess subpixel snow mapping performance while considering the fractional snow covered area (fSCA), canopy cover, sensor zenith angle, and other variables within six global seasonal snow classes. Metrics are calculated at the pixel and basin scales, including the root-mean-square error (RMSE), bias, and F statistic (a detection measure). The newer MOD10A1F Version 61 and VNP10A1F Version 1 product biases (-7.1%, -9.5%) improve significantly

when linear equations developed for older products are applied (2.8%, -2.7%) to convert band ratios to fSCA. The F statistics are unchanged (94.4%, 93.1%) and the VNP10A1F RMSE improves (18.6% to 15.7%) while the MOD10A1F RMSE worsens (12.7% to 13.7%). Consistent with previous studies, spectral mixture approaches (SCAG, SPIReS) show lower biases (-0.1%, -0.1%) and RMSE (12.1%,12.0%), with higher F statistics (95.6%, 96.1%) relative to the band ratio approaches for MODIS. Landsat 8 products are all spectral mixture methods with

low biases (-0.4 to 0.3%), low RMSE (11.4 to 15.8%), and high F statistics (97.3 to 99.1%). Spectral unmixing methods can improve snow cover mapping at the global scale.





## 1 Introduction

Snow cover is a globally significant climate forcing (Hansen and Nazarenko, 2004) and provides the water supply for billions of people (Mankin et al., 2015). Dramatic shifts in water availability are projected over the next 50 years due to climate change (Mankin et al., 2015; Immerzeel et al., 2020; Immerzeel et al., 2010) and resulting from regional disturbances that accelerate snowmelt timing (e.g., dust on snow; Deems et al., 2013b; Skiles et al., 2012). As regions warm, the fraction of precipitation that falls as rain, rather than as snow, increases, and glacial ice melts increasingly

rapidly and disappears (Stewart et al., 2005; Zemp et al., 2015). Understanding the recent trends in (Bormann et al., 2018) and trajectory of global snow cover is critical for comprehending global climate change and its impacts on the lives of billions of people.

      At the regional to global scales, satellite remote sensing is the best tool for measuring snow cover and snow albedo across landscapes (Molotch et al., 2004; Lettenmaier et al., 2015; Bair et al., 2019). The capability to map snow

cover from space was realized early in the era of spaceborne remote sensing (Dozier et al., 1981; Warren, 1982), and spaceborne multispectral instruments are now routinely used to monitor many snow surface properties: the fractional snow covered area (fSCA), snow albedo, snow grain size, reduction in albedo from light-absorbing particles (LAPs), and snow surface temperature (Painter et al., 2009; Painter et al., 2012; Lundquist et al., 2018; Bair et al., 2019; Nolin, 2010). Furthermore, remotely sensed snow cover information can be used to derive a variety of snow metrics that are

relevant to the changing climate and to hydrologic systems (Nolin et al., 2021). These metrics and snow surface properties have been used to estimate persistent ice cover (Painter et al., 2012), analyze the impacts of wildfire on snowmelt (Micheletty, 2014), evaluate continental climate models (Minder, 2016), force regional climate models (Oaida, 2019), partition snow and glacier melt (Armstrong et al., 2018), reconstruct snow water equivalent (SWE) (Guan, 2013; Bair et al., 2016; Rittger et al., 2016), quantify anthropogenic LAP impacts on snowmelt timing (Bair et

al., 2021b; Bair et al., 2016), and forecast streamflow (Micheletty, 2021).

      Fundamental to mapping snow with remote sensing is the knowledge that snow cover varies at a finer spatial scale than the scale of the data collected by most current and upcoming spaceborne optical sensors. Thus, relevant studies require subpixel retrievals. The importance of subpixel snow is well recognized, as evidenced by the development of fSCA approaches for application to high spatial resolution Airborne Visible/Infrared Imaging

Spectrometer (AVIRIS) data in the early 1990s and to high resolution commercial satellite data over the past decade (Nolin and Dozier, 1993; Selkowitz et al., 2014). At a 463 m spatial resolution (i.e., the resolution of the Moderate-resolution Imaging Spectroradiometer (MODIS)), many pixels are not fully snow covered, even in the middle of winter (Fig. 1). Even at a 30 m spatial resolution (i.e., that of Landsat 8), 25%-93% of pixels in mid latitude alpine environments are mixed pixels (Selkowitz et al., 2014). Though fully snow covered pixels are more common in some

relatively high latitude regions that contain extensive permanent snow and ice, e.g., Greenland, mixed pixels are pervasive at the boundaries of ice sheets and caps in these areas. Likewise, high latitude boreal forests contain mixed pixels. To estimate fSCA—a critical parameter in snow and terrestrial hydrology research—sub-pixel snow cover estimates are needed, as snow cover properties (e.g., albedo) differ from whole-pixel properties when pixels are not fully snow covered.



70        Satellites provide consistent global snow cover information, yet to date, no comprehensive validation of sub-pixel snow cover mapping has been conducted with independent data covering multiple snow climates. In this study, we evaluate and compare snow cover retrievals among multiple sensors and algorithms. High spatial resolution (3 m) snow depth data derived from airborne light detection and ranging (lidar) sensors, which can penetrate forest canopies, are used in this work to create snow cover maps to validate a suite of snow cover products.

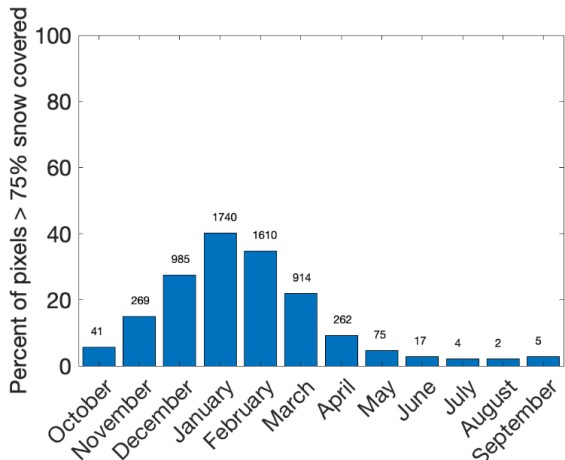

**Figure 1. Percent of all snow covered pixels with > 75% viewable fSCA in the Western US averaged from 2001 to 2021 from STC-MODSCAG (Rittger et al, 2020). The number above each bar represents the snow cover extent in km² averaged over the month of measurement. Pixels are 463 m.**

**2 Background**

We focus on validating snow mapping algorithms that determine fSCA using either empirical relationships based on the Normalized Difference Snow Index (NDSI, a band ratio technique) (Salomonson and Appel, 2004) or spectral mixture analysis (Nolin et al., 1993). Snow can be distinguished from other surfaces using NDSI (Eq. (1)) (Dozier, 1989) because snow is highly absorptive (low reflectance) in the shortwave infrared (SWIR) wavelengths and highly

reflective in the visible (VIS) wavelengths compared to most other land surfaces (Fig. 2):

$$NDSI = \frac{R_{VIS,\lambda} - R_{SWIR,\lambda}}{R_{VIS,\lambda} + R_{SWIR,\lambda}} \qquad (1)$$

where NDSI ranges from -1 to +1 and $R$ is the reflectance for each band in the subscript. The simplicity of NDSI is appealing and explains its prevalence (Hall et al., 2002; Salomonson and Appel, 2004; Hall et al., 2010; Justice et al., 2013). However, NDSI conveys no information regarding the spectral signature of snow; various non-snow mixed pixels can yield the same NDSI values as snow covered pixels (Stillinger et al., 2019). Additionally, NDSI is often

used with a threshold to create binary snow cover maps. However, these thresholds have been shown to be both spatially and temporally nonstationary (Harer et al., 2018; Tong et al., 2020). Further, the uncertainty of these thresholds increases with an increased spatial resolution (e.g., those of Landsat satellites) and in forested areas (Klein et al., 1998). Past studies have shown that NDSI is less accurate than spectral unmixing techniques when estimating

fSCA over various terrains and at both middle and high latitudes (Rittger et al., 2013; Masson et al., 2018; Aalstad et

al., 2020).

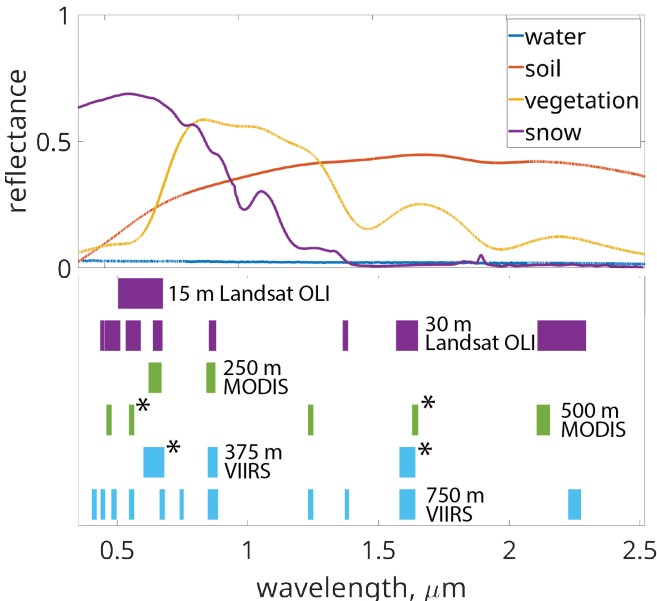

**Figure 2. Hyperspectral measurements of common land surfaces: average alfisol soil, *Pinus* vegetation, and water spectra (Meerdink et al., 2019; Baldridge et al., 2009) along with a typical dirty mountain snowpack from Mammoth Mountain, CA on 01 May 2021. The lower panel shows the bands and spatial resolutions corresponding to the satellite sensors used in this study. The black asterisks denote the NDSI bands, while the spectral unmixing approaches use all bands to map snow.**

A more sophisticated and physically based method, spectral mixture analysis, is an inversion approach derived for various surfaces; in this method, the measured reflectance is matched to a modeled reflectance to estimate endmember fractions and surface properties (Adams et al., 1986; Roberts et al., 1998). Multiple linear equations are simultaneously solved for endmembers (Eq. (2)), and $R_{s,\lambda}$ is the modeled surface reflectance at wavelength λ:

$$R_{s,\lambda} = \sum_{i=1}^{N} F_i R_{\lambda,i} + \varepsilon_\lambda \qquad (2)$$

where $F_i$ is the fraction of endmember $i$; $R_{\lambda,i}$ is the reflectance of endmember $i$ at wavelength $\lambda$; N is the number of endmembers; and $\varepsilon_\lambda$ is the residual error. The model is run iteratively to minimize the root-mean-square error (RMSE) between the modeled and observed surface reflectance values. This approach was first used to estimate snow properties from hyperspectral AVIRIS airborne data (Nolin and Dozier, 1993; Nolin et al., 1993; Painter et al., 1998; Painter et al., 2003). The MODIS Snow Covered Area and Grain size (MODSCAG–or SCAG more generally) algorithm (Painter

et al., 2009) proved that spectral unmixing was a viable approach to snow mapping when using multispectral satellite data. The MODSCAG algorithm can map fSCA at larger spatial scales and with more frequent temporal repeats than those possible with airborne AVIRIS-derived hyperspectral datasets. MODSCAG solves for each pixel in each image



and, in addition to fSCA, also outputs fractional vegetation and fractional soil and rock information. A modified version of the SCAG algorithm (Rittger et al., 2021a) is used by the United States Geological Survey (USGS) in their

fSCA Landsat product (Selkowitz et al., 2014). Developed more recently, the Snow Property Inversion from Remote Sensing (SPIReS) (Bair et al., 2021a) spectral unmixing approach builds on the previously described spectral unmixing efforts but uses only two endmembers (snow and snow-free) plus ideal shade to simultaneously solve for fSCA, the snow grain size, and the snow contaminant concentration. Other research-based approaches to spectral mixture analysis have been developed for multispectral satellite data, such as the MODIS Imagery Laboratory

(MODiMlab) (Sirguey et al., 2009) or *SnowFrac* (Vikhamar and Solberg, 2003), but these approaches are not available for the dates or areas considered in this analysis.

Despite the long history of mapping snow from space, spatial validations of the available standard (produced operationally by an agency) products are limited. Shortly after MODIS-derived standard snow products were released in the early 2000s, researchers compared sparse subsets of individual pixels from these new standard snow products

to point-scale snow depth measurements (Simic et al., 2004; Ault et al., 2006; Klein and Barnett, 2003; Maurer et al., 2003). These comparisons were followed by fSCA validations performed using relatively coarse resolution datasets, e.g. 463 m pixels compared to 250 m pixels in Hall and Riggs (2007). More extensive validations were performed for MOD10A1 binary snow cover, MOD10A1 fSCA, and MODSCAG (Rittger et al., 2013). Using 172 scenes, Rittger et al. (2013) assessed the accuracies of these products in a number of regions across the Western US and in the Himalaya

by applying spectral mixture analysis to the Landsat Enhanced Thematic Mapper Plus (ETM+) surface reflectance product at a 30 m scale. The results showed improvements resulting from using a fractional approach at 30 m compared to a binary approach, especially when performing validations in forested regions.

These past studies assessed only the viewable snow cover, i.e., the snow cover that is directly measurable from space with optical sensors during cloud-free overpasses. However, in forested regions, canopies obstruct snow

cover, making validations under forest canopies difficult. Reference validation data are typically derived either from ground stations (typically representing point data in open areas) or from Landsat (limited to viewable fSCA). Recent studies have provided insights into snow cover mapping in forests using novel ground-based data and airborne lidar measurements. Raleigh et al. (2013) used gridded soil temperature networks to evaluate the MODSCAG algorithm using a static canopy adjustment method against sites exhibiting a range of canopy cover in the Sierra Nevada. Rittger

et al. (2020) used the same dataset to validate a viewable snow cover to on-the-ground snow cover adjustment process involving the simultaneously retrieved pixel vegetation fraction in SCAG for pixels with canopy cover fractions up to 0.75. Bair et al. (2021a) compared MODIS- and Landsat-derived fSCA data to Worldview 2/3 and Airborne Snow Observatory aerial lidar data (Painter et al., 2016) while focusing their validation on the difference between the viewable fSCA (Worldview validation) and canopy-corrected fSCA (the ground snow cover validated with aerial

lidar).

While these past studies advanced validation efforts, the present study offers two notable improvements: 1) all currently available snow mapping products (including new gap-filled products) are compared across 2) a diverse range of snow climates. With the emergence of new snow mapping products and available high-spatial-resolution (i.e.,



meter scale) snow depth estimates derived from aerial lidar technology, the comprehensive evaluation undertaken herein is critical.

## 3 Study area

Our study area comprised regions in California and Colorado in the Western US (Table 1). Validation locations were selected based on the availability of Airborne Snow Observatory (ASO) (Painter et al., 2016) snow depth measurements obtained during snow-covered and snow-free flights using a lidar instrument (Sect. 4.2.1). Full-
waveform lidar instruments are usually able to penetrate forest canopies, representing a significant advancement in aerial snow mapping (Deems et al., 2013b; Deems et al., 2013a). Figure 3 shows the spatial extents of the validation regions and the Sturm and Liston (2021) snow type classification scheme (Sect. 4.2.4) corresponding to each region. Quantifying algorithm performance from the perspective of snow types enables a better understanding of expected performance in geographical regions without validation data or past studies. Additionally, fSCA can depend on the
snow climate (Liston, 2004; Clark et al., 2011); thus, validating across various snow climates allows a range of sub-pixel snow distributions and depletion dynamics to be sampled, thereby strengthening our confidence in the global application abilities of these products.

**Table 1. Watershed and snow type characteristics characterizing each study location.**

| Watershed name | State | ASO flights (#) | Watershed area (km$^2$) | Canopy cover | Snow class fraction | | | | | |
| --- | --- | --- | --- | --- | --- | --- | --- | --- | --- | --- |
| | | | | | Tundra | Boreal forest | Maritime | Ephemeral | Prairie | Montane forest |
| Kings Canyon | California | 13 | 3565 | 25% | 7% | 0% | 19% | 12% | 24% | 39% |
| Merced River | California | 8 | 835 | 20% | 6% | 0% | 55% | 2% | 15% | 21% |
| San Joaquin | California | 21 | 4234 | 29% | 6% | 0% | 35% | 20% | 14% | 26% |
| Tuolumne River | California | 34 | 1674 | 12% | 6% | 0% | 35% | 1% | 30% | 29% |
| Kaweah River | California | 3 | 1450 | 37% | 0% | 0% | 14% | 40% | 9% | 37% |
| Lakes Basin | California | 14 | 28 | 16% | 19% | 0% | 0% | 0% | 20% | 60% |
| Lee Vining Creek | California | 1 | 114 | 8% | 28% | 2% | 0% | 0% | 28% | 41% |
| Rush Creek | California | 1 | 139 | 9% | 24% | 1% | 0% | 0% | 27% | 49% |
| Reds Lake | California | 1 | 2 | 20% | 1% | 0% | 0% | 0% | 38% | 61% |
| Blue River | Colorado | 4 | 866 | 25% | 31% | 44% | 0% | 0% | 6% | 19% |
| Crested Butte | Colorado | 2 | 178 | 18% | 31% | 21% | 0% | 0% | 6% | 42% |
| Conejos River | Colorado | 1 | 729 | 27% | 13% | 16% | 0% | 0% | 13% | 58% |
| Aspen/Castle-Maroon | Colorado | 2 | 326 | 22% | 47% | 27% | 0% | 0% | 1% | 24% |
| Gunnison-East River | Colorado | 2 | 1670 | 24% | 21% | 24% | 0% | 0% | 13% | 41% |
| Grand Mesa | Colorado | 5 | 322 | 29% | 6% | 64% | 0% | 0% | 1% | 29% |
| Gunnison-West River | Colorado | 3 | 658 | 29% | 22% | 38% | 0% | 0% | 6% | 34% |
| Rio Grande River | Colorado | 1 | 1862 | 16% | 25% | 32% | 0% | 0% | 17% | 27% |

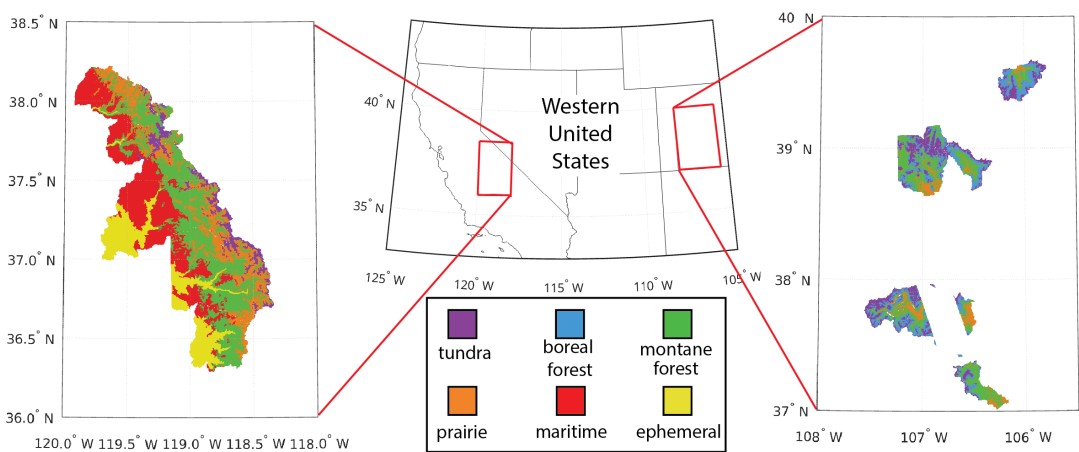


**Figure 3. Snow cover types present in the ASO validation flight locations. All six of the (Sturm and Liston, 2021) seasonal snow type classes are present in the combined California and Colorado ASO datasets.**

## 4 Methods

Table 2 lists the seven snow cover products evaluated against snow cover maps derived from high-resolution airborne
lidar data. These products include two NDSI-based standard products, MOD10A1F and VNP10A1F, and five spectral
mixture products, the USGS Collection 1 Landsat 8 Level-3 fSCA product (USGS FSCA), the Snow-Covered Area
and Grain size (SCAG) algorithm run on MODIS (STC-MODSCAG) and Landsat 8 (OLISCAG), and the SPIReS
algorithm run on MODIS and Landsat 8. The daily products are all gap-filled products. All products are described in
detail in individual sections below.

**Table 2. Snow cover products validated in this study and their characteristics.**

| Product | Resolution (temporal / spatial) | Reference | Canopy adjustment method | Minimum snow cover fraction (besides 0) | Gap-filling method | Near real time |
|---|---|---|---|---|---|---|
| MOD10A1F | Daily / 463 m | Hall et al. (2019) | None | 0.1 NDSI snow cover; 0.135 fSCA | Hall et al. (2010) | Yes |
| VNP10A1F | Daily / 375 m | Riggs et al. (2019) | None | 0.1 NDSI snow cover; 0.135 fSCA | Hall et al. (2010) | Yes |
| USGS FSCA | 16 days / 30 m | Selkowitz et al. (2017) | Spatial replacemnet Selkowitz et al. (2017); (Sect. 4.1.5) | 0.15 fSCA | None | Yes |
| STC-MODSCAG | Daily / 463 m | Painter et al. (2009); Rittger et al. (2020) | Rittger et al. (2020); scaled adjustment (Sect. 4.1.3) | None | Rittger et al. (2020) | Yes |
| OLISCAG | 16 days / 30 m | Rittger, Bormann, et al. (2021) | Rittger et al. (2020); scaled adjustment (Sect. 4.1.4) | 0.1 fSCA | None | No |
| SPIReS (MODIS) | Daily / 463 m | Bair et al. (2021a) | Viewable gap fraction (Sect. 4.1.6) | 0.1 fSCA | Bair et al. (2021a) | No |
| SPIReS (Landsat 8) | 16 days / 30 m | Bair et al. (2021a) | Scaled adjustment and spatial interpolation (Sect. 4.1.7) | 0.1 fSCA | None | No |





We selected data from all products that matched the dates of ASO flights from 2013–2020. On 20 days, an ASO flight occurred on the same day as a Landsat 8 data acquisition. Three of the ASO flights corresponded with extensive thin cloud cover during the corresponding Landsat 8 overpass (although not during the ASO overpass) and were thus removed to validate snow mapping abilities under clear skies. Other scenes contained small amounts of cloud cover. These pixels were masked and removed.

Because the MODIS and the Visible Infrared Imaging Radiometer Suite (VIIRS) products have daily temporal resolutions, corresponding MODIS/VIIRS snow cover maps are available for all ASO flight dates. ASO sometimes mapped multiple watersheds on the same day, and these products can be included within the bounds of a single MODIS/VIIRS tile. For STC-MODSCAG and MODIS SPIReS there are 116 ASO flights. The historical records of MOD10A1F and VNP10A1F have not been processed completely (as of May 2022, the time of the analysis), so we used all currently available products, totaling 87 ASO flights for VNP10A1F and MOD10A1F. These validation data provided sufficient coverage for conducting comprehensive analyses of MOD10A1F and VNP10A1F. For MODIS and VIIRS, all validation scenes were used because all algorithms used to construct these products include temporal filters that improve the snow mapping ability based on data collected on adjacent days (a feature that no Landsat 8 product has). Including all scenes thus supported the goal of this work regarding the validation of daily gap-filled products. The standard VIIRS product, VNP10A1F, was the only VIIRS product examined. Both SCAG and SPIReS can run on VIIRS but have not yet been produced for the dates and areas considered in this study. We note that VIIRSCAG shows a nearly identical performance to that of MODSCAG (Rittger et al., 2021a).

## 4.1 Validated snow cover products

### 4.1.1 MOD10A1F

From the suite of MOD10 snow products, all of which use the same snow detection algorithm, MOD10A1F was selected; this product is the currently available gap-filled Collection 6 MODIS snow product. MOD10A1F is an NDSI-based snow cover product in which NDSI values are delivered as snow cover estimates; NDSI values below 0.1 are flagged as no snow. The product is constructed daily at a 463 m spatial resolution and undergoes gap-filling to adjust for cloud cover or poor data by reusing measurements taken on the most recent clear-sky day. This product is not always spatially or temporally complete as there is no backwards interpolation of snow cover if the initial days of the water year are cloud covered. However, NDSI cannot be used as a direct estimate of the fSCA value of a given pixel. In the deprecated MODIS MOD10A Collection 5 products, additional processing was conducted to convert NDSI values to fSCA values; however, these products are no longer delivered. In addition to comparing the Collection 6 NDSI snow cover information contained in MOD10A1F, we estimated fSCA using the Collection 5 MODIS Terra relationship (Eq. (3)). This relationship was originally developed using Landsat data, and we assumed here that the equation parameters and formulation are still valid for the Collection 6 products (Salomonson and Appel, 2006, 2004). Notably, with the applied correction, the minimum possible fSCA (besides 0) value is 0.135.

$$fSCA = -0.01 + (1.45 \times NDSI) \tag{3}$$



Previous MOD10 collections included processing steps in which the normalized difference vegetation index (NDVI) was used to improve the snow detection ability in forested areas (Klein et al., 1998); however, low NDSI values should be mapped as snow using the updated fractional method (Salomonson and Appel, 2004).

### 4.1.2 VNP10A1F

We accessed the first version of the VIIRS standard snow product, VNP10A1F (Riggs et al., 2019); this product is nearly identical to MOD10A1F, but VIIRS obtains surface reflectance data at slightly different bandpasses and spatial resolutions (see Fig. 1). All VNP10 snow products use the same snow detection algorithm as the MOD10 products. The utilized product was a 375 m resolution daily snow product delivered as a gap-filled NDSI product. The same gap-filling approach used to construct MOD10A1F was applied to VNP10A1F. As no published adjustments are

available for converting NDSI values to fSCA values, we also reused the same retrieval algorithm applied to MOD10A1F, shown in Eq. (3).

### 4.1.3 STC-MODSCAG

MODSCAG (Painter et al., 2009) is a spectral mixture analysis approach that uses a library of endmembers representing clean snow reflectance, soil/rock, vegetation, and photometric shade and performs least-squares fitting

to obtain fSCA from MODIS Terra surface reflectance products (MOD09GA). In this approach, the fraction of each of the endmembers was estimated along with the snow grain size corresponding to the snow endmember for each pixel in every image. Some non-linear effects were incorporated by including canopy-level vegetation endmembers. The data were then gap-filled based on spectral and persistence filters (Rittger et al., 2020) and interpolation while preferentially weighting nadir views over off-nadir views (Dozier et al., 2008). This gap-filled product is called the

Spatially and Temporally Complete MODSCAG (STC-MODSCAG) product. Viewable fSCA, fractional vegetation (fVEG), and fractional soil/rock (fROCK) information available from MODSCAG is used for STC-MODSCAG. In STC-MODSCAG, vegetation height maps (Simard et al., 2011), viewable fSCA, and fVEG are used to estimate snow on the ground. When the viewable fSCA is greater than 0.1 in forested areas identified by a vegetation height greater than 2.5 m, fSCA is adjusted using concurrently observed fVEG. Rittger et al. (2020) showed the STC-MODSCAG

canopy correction method to be a function of the view angle. Additional post processing was conducted to increase the accuracy of the results in both forested and alpine regions. In forested areas, the maximum snow cover is limited using a vertical-to-horizontal crown radius ratio (Liu et al., 2004) to remove snow where tree trunks exist. In alpine areas (i.e., areas with vegetation heights < 2.5 m and elevation > 800 m in this analysis), the retrieved fSCA was scaled by +10% of the initial value. The data utilized in the analyses performed in this paper were produced by Snow Today

at the National Snow and Ice Data Center (NSIDC, https://nsidc.org/snow-today) (Rittger and Raleigh, 2022). Snow Today is supported by NSIDC Distributed Active Archive Center (DAAC) User Services and has provided freely available daily images in near real-time since 2020. The historical records from earlier STC-MODSCAG versions are available via file transfer protocol (FTP) at snow.ucsb.edu.



### 4.1.4 OLISCAG

OLISCAG uses the same spectral mixture analysis approach as MODSCAG but applies this approach to Landsat 8 Operational Land Imager (OLI) data (Rittger et al., 2021a). The spectral libraries differ based on the number and size of bandpasses used by each instrument. The images are not interpolated temporally or spatially as is performed with the STC-MODSCAG data (described above). The SCAG model was generalized for the Landsat ETM+ and TM instruments (Rosenthal and Dozier, 1996; Painter et al., 2009; Rittger et al., 2013), and Rittger et al. (2021a) modified

the SCAG algorithm to work with the new 12-bit OLI data using spectral libraries updated specifically for OLISCAG. Similar to STC-MODSCAG, a canopy adjustment is applied to the OLISCAG viewable fSCA to obtain on-the-ground fSCA. The viewable fSCA is adjusted when a minimum fSCA is detected (0.1) using the concurrently estimated vegetation fraction and the crown ratio equation in Liu et al. (2004), similar to the processing methods described above for STC-MODSCAG. OLISCAG has a minimum fSCA detection threshold set at 0.1.

### 4.1.5 USGS Landsat fSCA

fSCA is available from the Landsat 8 Collection 1 Level-3 products provided by the USGS. The Collection 1 product is produced for the Western US and Alaska in the Landsat Analysis Ready Data tiles and is constructed through a spectral mixture analysis based on the SCAG algorithm (Painter et al, 2009) using spectral libraries from OLISCAG (Rittger et al., 2021a). This product represents a significant advancement over the standard snow maps available in

the Level 1 and 2 band quality assessment files delivered with Landsat 8 reflectance data. The SCAG implementation includes the application of snow-specific cloud-masking, water-masking, and canopy cover adjustments to construct the final fSCA product (Selkowitz et al., 2017). The primary difference between OLISCAG and the USGS fSCA product is in their canopy correction approaches. In the USGS fSCA product, snow cover in pixels identified as forest pixels using the National Land Cover Database (NLCD) (Homer et al., 2004) is replaced with snow cover from less

forested nearby pixels with similar accumulated solar radiation and elevation. This product has a minimum fSCA detection threshold set at 0.15.

### 4.1.6 MODIS SPIReS

SPIReS (Bair et al., 2021a) is an open-source (see Sect. 8) spectral unmixing approach designed to map fSCA using pixels from snow-free periods as background reflectance information and a modeled snow endmember that contains

contaminants (e.g. dust or soot). These two endmembers are mixtures themselves, thus reducing the number of unknowns and making the system more highly overdetermined (Branham, 1990), in turn increasing the accuracy and computational speed. Additionally, the background endmember mixtures account for some pixel-specific non-linear effects, such as canopy effects. SPIReS clusters pixels within a tolerance, then simultaneously solves for fSCA, the grain size, and the contaminate concentration at the surface of the snowpack for each cluster and applies the solution

to all pixels in each cluster. This process typically results in a 10-100× reduction in the number of unmixing model runs required compared to solving for every pixel. The latter two properties are then used to estimate snow albedo (Bair et al., 2019). SPIReS adjusts the viewable snow cover to account for shading, permanent ice, and snow hidden by forest canopies. Following the initial publication of the SPIReS method by Bair et al. (2021a), SPIReS was adjusted



to better handle specific snow mapping situations. If a pixel is fully snow covered, no snow-free background surfaces
should be visible. SPIReS accounts for this issue by solving for pixels twice, first with the standard method including
a "snow-free" background reflectance and then again with no background. If this second solution is within 2% of the
original solution, the new "no-background" solution is used. The goal of this double-solving approach is to
significantly reduce the biases that arise for pixels with high fSCA values.

In MODIS SPIReS, the canopy correction step is based on the viewable gap fraction (VGF) estimated for
each pixel, i.e., an estimate of the fraction of the pixel that comprises viewable ground. This fraction is a function of
the canopy cover, topography and satellite view angle. VGF adjustments are performed when constructing MODIS
SPIReS to account for the satellite view angle based on the Geometrical Optical Model (Liu et al., 2004). For MODIS
SPIReS, persistence filters based on both the snow cover and minimum snow grain size alongside temporal smoothing
with weighted splines (Dozier et al., 2008) are used to generate daily, gap-free snow cover and surface snow property
estimates corresponding to each pixel. While Bair et al. (2021a) found through an F statistic analysis that the fSCA
detection abilities were maximized at 0.01 for MODIS and 0.07 for Landsat OLI, the minimum fSCA threshold for
SPIReS was set to 0.1 in this study, the current default setting.

### 4.1.7 Landsat SPIReS

Landsat SPIReS is constructed by applying the same approach as MODIS SPIReS, with a few differences. Because
Landsat is a nadir-looking pushbroom instrument, the Geometrical Optical Model is not used. Instead of a VGF
adjustment, for pixels with canopy cover fractions above 0.5, fSCA is ignored and spatial interpolation is used to
spread snow into these pixels in which the direct snow cover measurements are unreliable. Similar to the OLISCAG
and USGS fSCA products, no gap-filling procedures (apart from the canopy correction step described above) are used
to construct the Landsat SPIReS product.

## 4.2 Validation datasets

### 4.2.1 ASO snow depth

ASO produces snow depth and SWE products that are archived at NSIDC (2013–2019) and at the ASO, Inc. website
(2019–present) for use by researchers and water managers (Painter et al., 2016). We obtained all available ASO data
from water years 2013–2020. In this study, the 3 m snow depth product was first converted into a binary snow map
and then coarsened into an fSCA product matching the spatial resolution and projection of MODIS, VIIRS, or Landsat
(as described below); these fSCA products were then used to validate the satellite-derived snow products (Sect. 4.3).

The 3 m ASO snow depth data represent an intermediate ASO product without the same level of quality
control as the final 50 m SWE products. The 3 m datasets are associated with extensive data representation issues; for
example, "zero" is used as both a fill value and a snow-free classifier. To convert the 3 m snow depth data to high-
quality 3 m snow cover maps, additional data cleaning was thus mandatory. To improve the quality of the 3 m snow
depth data and to ensure the use of valid measurements, the data extent was limited to the basin boundaries and to the
3 m depth measurements co-located with valid measurements in the 50 m SWE dataset derived during the same flights.
The validation datasets were cropped to the basin boundaries even when the flight lines extended past these



boundaries, as the data collected outside the basin boundaries were often unreliable. At certain times late in the melt
season, when snow was present only at the highest elevations in the studied basins, flights did not extend across the
whole basins; under these conditions, we used the 50 m SWE datasets with snow-free, snow, and missing-value
representations to select valid spatial regions from the 3 m snow depth products.

To convert the ASO snow depths to fSCA maps, the cleaned ASO-derived 3 m snow depth data were first
relabeled as snow-free, snow-covered, or filled data. The mean absolute error (MAE) of the 3 m snow depth product
has been reported to be less than 8 cm (Painter et al., 2016); hence, these snow depth measurements were converted
into a 3 m binary snow cover mask in which the snow cover was considered true when the snow depth was greater
than 8 cm, "snow free" areas referred to measured snow depths of 8 cm or less, and "fill" data were established where
either the 3 m depth product had missing data, the 50 m SWE product had missing data, or the analyzed pixels were
located outside the basin boundaries. These binary 3 m snow cover maps were then coarsened to the same spatial
resolution as each fSCA satellite product for the subsequent validation. The ASO-derived snow cover validation
datasets were reprojected into the projection and spatial grid of each satellite product. Any reprojected validation
pixels with fSCA values less than 0.01 were assumed to be snow-free (i.e., fSCA=0). While these threshold choices
will impact the validation results, the goal of the chosen thresholds was to eliminate artifacts resulting from upscaling
the 3 m data to coarser resolutions (i.e., 120 m for Landsat and 2 km for MODIS/VIIRS validation; Sect. 4.3.1) while
comparing as many pixels as possible to avoid ignoring low snow covered pixels, as was done in older (Painter et al.,
2009; Rittger et al., 2013) but not newer (Masson et al., 2018) studies. Additionally, a goal of this work was to reduce
the probability of false positive snow cover measurements arising due to the presence of thin snow measured within
the uncertainty range of the ASO lidar data. ASO, like any validation source, is imperfect. ASO may not map all rock
outcrops correctly in alpine regions and may instead consider those locations to be fully snow covered. Additionally,
in regions with steep terrain and dense forests, depending on the orientation of the flight line relative to the underlying
surface, the lidar retrieval quality may vary. Even given these concerns, a comparison between ASO and Worldview
2/3 high-resolution optical imagery derived in snowy mountainous terrain has shown that ASO is a high-quality
validation source (Bair et al., 2016; Bair et al., 2021a), and we treated these data as validation truth in this paper.

### 4.2.2 Canopy cover

The static NLCD 2016 tree canopy cover dataset (Wickham et al., 2021) with a 30 m spatial resolution was used to
determine the canopy cover locations and fractions (0-1) within the validation regions. NLCD 2016 uses spectral and
geographical information to determine land cover types and was chosen due to its operational availability and 30 m
spatial resolution. These NLCD canopy cover files were cropped and stored with the ASO dataset representing the
corresponding day and then reprojected alongside the ASO validation datasets to the native projection and resolution
of each product. These coarsened canopy cover maps were then used to categorize pixels into canopy cover bins for
the subsequent statistical analysis across the full range of existing canopy cover fractions in the study areas.





### 4.2.3 View angle

MODIS and VIIRS are both scanning whiskbroom sensors that have sufficiently wide fields of view with variations in the sizes of individual pixels across a given swath (Dozier et al., 2008). In addition to these pixel size variations, off-nadir views, especially those obtained in pixels with canopy cover, cause variations in the fraction of the surface under the canopy that is visible to the satellite in each pixel (Rittger et al., 2020; Bair et al., 2021a). For both MODIS and VIIRS, NDSI-based approaches do not adjust for the view angle, whereas the SCAG and SPIReS approaches do account for the view angle (see above). To understand how well these algorithms map snow cover when high-view-angle acquisitions have occurred, we took the per-pixel satellite view zenith angle information from the top layer in the MOD09GA or VNP09GA surface reflectance product corresponding to the date and location of the target snow cover estimate and binned the derived statistical results by the viewing geometry in 5° bins ranging from 0° (nadir) to 70° (edge of scan). As Landsat carries a pushbroom sensor, it acquires near-nadir images across all pixels, so no view angle analyses were performed for the Landsat 8 datasets.

### 4.2.4 Global seasonal snow classifications

The study area considered herein was classified into seasonal snow cover types using the Sturm and Liston (2021) classification scheme, which includes 6 snow categories (tundra, boreal forest, maritime, ephemeral, prairie, and montane forest snow) on a 1 km global classification map. These data are available at the NSIDC website (https://nsidc.org/data/NSIDC-0768/versions/1). For each product, the derived error statistics were binned into the snow class corresponding to each fSCA pixel using a nearest-neighbor resampling approach.

### 4.3 Validation

### 4.3.1 Upscaling

In the validation process, the ground-truth snow cover maps derived from ASO were compared to each fSCA product. The abilities of each algorithm to detect snow (Sect. 4.3.1) and correctly estimate the fSCA values in each pixel and in each watershed basin (Sect. 4.3.2) were evaluated separately. To account for geolocational uncertainty in the satellite products, we upscaled the fSCA values obtained from each product before the validation. The Landsat 8 products were validated at a 120 m spatial resolution, and the MODIS/VIIRS products were validated at a 2 km spatial resolution. Both the ASO validation data and the satellite data were coarsened to the spatial resolution and original projection of the corresponding satellite products. This same approach has been adopted in past validation studies in which snow products were compared to relatively high-spatial-resolution validation data (Stillinger et al., 2019; Bair et al., 2021a; Bair et al., 2016; Rittger et al., 2013; Rittger et al., 2020). While VIIRS data are available at a 375 m spatial resolution compared to the 463 m resolution of MODIS data, both products were validated herein at a 2 km spatial resolution. Each product is validated as they are delivered to users; thus, no additional thresholds were applied to set a minimum snow cover.



### 4.3.2 Snow detection

To validate the snow detection results, a binary mask of ASO-derived snow cover (ASO fSCA>0) was compared to binary masks of the product-derived snow cover (product fSCA >0), and snow cover was considered a true positive (TP) in all cases where fSCA>0. Four pixel classes were generated from the comparison: TPs, false positives (FPs), true negatives (TNs), and false negatives (FNs). TNs and, subsequently, the commonly used "accuracy" statistic that includes TNs were not used in the subsequent validation assessments as they can skew the results when large swaths

of easily classified snow-free areas exist in an image. From the remaining three pixel classes (TPs, FPs, and FNs), the precision, recall, and F statistic values were calculated (Eqs. (4), (5) and (6)).

$$Precision = \frac{TP}{(TP + FP)}$$
(4)

$$Recall = \frac{TP}{(TP + FN)}$$
(5)

$$F = \frac{(2 \times precision \times recall)}{(precision + recall)}$$
(6)

   Precision is a measure of how few FPs a product generates, characterizing the ability of an algorithm to include only snow in its snow classification results. A high precision of 99% means that 99% of the pixels mapped as snow were actually snow-covered according to the ground-truth data. Recall is a measure of how few FNs a product

generates, characterizing the ability of the algorithm to map all the snow in an image. A high recall of 99% means that 99% of the snow-covered pixels in an image were included in the snow mask produced by the algorithm. The F statistic, the harmonic mean of precision and recall, is a way to balance these two independent algorithm performance metrics. A high F statistic means that the algorithm correctly mapped the snow in an image and did not include other land surface types in the produced snow maps.

**4.3.3 Per-pixel and per-basin fSCA**

   The upscaled ASO fractional snow cover validation data discussed in Sect. 4.3.1 were compared to the fSCA estimates obtained from each product at the given validation-step spatial resolution. The evaluation approach described in Sect. 4.3.2 provides a measure of the snow detection ability of an algorithm but does not measure the ability of the algorithm to correctly estimate fSCA in a pixel. A product-derived fSCA estimate of 0.2 compared to a true fSCA estimate of

0.95 would result in a TP snow detection but would suggest a large bias and error in terms of the accuracy of the fSCA estimate. To determine how accurate the analyzed algorithms are in terms of their produced fSCA estimates, the bias and RMSE values were calculated to evaluate how well each product performed when estimating fSCA at both the pixel and basin scales. Bias is the average overall difference in the fractional snow cover estimate between a product and the ASO snow cover fraction truth map at the analysis resolution. RMSE is a measure of the average error in



individual-pixel fractional snow cover estimates compared to the ASO snow cover fraction truth map at the analysis resolution.

The hydrological boundaries of basins were used when planning the ASO flights from which validation data were collected; assessing the snow cover mapping performance at this scale enables us to evaluate the ability of each product to calculate the total fSCA of a watershed. At the basin scale, the snow fraction of all validation pixels (TPs, 400    FPs, FNs) in each basin was calculated for each product and the ASO snow cover truth map by summing the per-pixel fractional cover estimates and dividing by the number of validation pixels. The snow products were then compared to the same measurement calculated from the ASO data. The basin-wide product and ASO-derived, basin-scale fSCA were compared in each basin on each day. Past error analyses have focused on per-pixel errors, which can be used to improve confidence when fSCA data are used for high-resolution, spatially explicit models working to better 405    understand processes. The new basin-wide error estimates derived herein provide information for larger-scale, coarser models like those used in global and regional climate simulations.

### 4.3.4 Specific pixel subcategories

Understanding the snow mapping performances of various algorithms under specific conditions is crucial for building confidence in algorithm quality and providing insight into scenarios that require further development to improve snow 410    mapping abilities. In this work, the pixels in each product were separated into categories, and the statistics derived across gradients in each category were evaluated. The pixels in the Landsat 8 and MODIS/VIIRS datasets were grouped by seasonal snow class and canopy cover fraction, and the MODIS/VIIRS dataset pixels were additionally grouped by satellite view angle. Additionally, visual representations of the error patterns were generated (see Fig. 7) by mapping the combination of the per-pixel snow cover bias from -99% to +99% alongside the TNs, replacing the - 415    100% bias measurements with FNs and replacing the +100% bias measurements with FPs. These errors, when displayed alongside ASO fSCA maps (representing the truth) and canopy cover maps, help deliver insights into the performances of the analyzed algorithms.

### 5 Results

### 5.1 Overall statistical assessment

Table 3 highlights the results obtained from the snow detection validation analysis performed while applying the ASO data as the truth dataset. The per-pixel fSCA and basin-wide fSCA results were assessed for each product in terms of the bias, RMSE, precision, recall, and F statistic metrics. The per-scene median, minimum, and maximum values corresponding to each statistical measure and to each product are listed in Table 3. Notably, the minimum bias and minimum recall statistics for a single product can be obtained on two different dates.

STC-MODSCAG, OLISCAG, the USGS FSCA, SPIReS Landsat and SPIReS MODIS all had median biases of ~0%. The standard snow mapping products from MOD10 and VNP10 had biases of -7.1% and -9.5%, respectively, but these biases were reduced to +2.8% and -2.7% when the Collection-5 fractional snow cover correction (Eq. (3)) was applied to the standard Collection 6 products, a step not taken in the archived data at NSIDC. When multiple



products have no bias, the product with the lower RMSE can be considered the better-performing product. The median

RMSE values (Table 3) derived herein ranged from 11.4% (OLISCAG) to 19% (VNP101AF).

The precision, recall, and F statistic metrics measure the ability of the analyzed products to detect snow cover (Table 3). The median precision values derived herein ranged from 89.4% (VNP101AF) to 99.4% (SPIReS Landsat). In this work, the median recall values ranged from 96.2% (MOD10A1F) to 100% (SPIReS MODIS and STC-MODSCAG). For Landsat products, the F statistic ranged from 97.3% (USGS) to 99.1% (SPIReS Landsat and

OLISCAG), while for MODIS and VIIRS, the F statistics were lower, ranging from 93.1% (VNP101AF) to 96.1% (SPIReS MODIS). For all snow cover detection measures, the algorithm performances associated with the worst scene exhibited wide variabilities; the minimum F statistic values ranged from 83.2% (USGS fSCA) to 0% (VNP10A1F).

The basin-wide fSCA RMSE and bias validation results are presented at the bottom of Table 3. OLISCAG, USGS Landsat, SPIReS Landsat, and SPIReS MODIS had overall basin-wide biases of ~0%. The standard NDSI

snow cover products (MOD10A1F and VNP10A1F) had higher overall biases of -9.2% and -11.9%, respectively. The RMSE values obtained at the per-basin scale were lower than those obtained at the per-pixel scale, as the overestimates and underestimates balanced each other in the basin-wide summation. The median RMSE values derived at the basin scale ranged from 4.7% (SPIReS Landsat) to 15% (VNP101AF).

When the Collection 6 MOD10A1F and Collection 1 VNP10A1F NDSI snow cover products were corrected

with the Collection 5 fractional cover linear correction (Eq. (3)), the precision, recall, and F statistic metrics were unchanged as the snow detection ability was unchanged, but the bias and RMSE values improved for both products at the per-pixel and basin scales as the fractional values were adjusted by the linear equation. All spectral unmixing methods showed comparable snow-identification performances (i.e., similar precision, recall, and F statistic values) across the satellite platforms and outperformed the NDSI products. The per-pixel bias and RMSE values were

comparable among all Landsat 8 products. Among the MODIS products, SPIReS and STC-MODSCAG exhibited lower median bias (~0%) and RMSE (12%) values than the standard MOD10A1F product.

**Table 3. Snow cover fraction product validation statistics obtained for all products assessed in this study along with the total and median per-scene counts of validation pixels at the validation resolution (120 m for Landsat, 2 km for MODIS/VIIRS) for all products. Note "FSC" denotes where Eq. (3) was used to estimate fSCA from operational NDSI**
**products.**





| | | Landsat 8 | | | MODIS | | | | VIIRS | |
| | | SPIReS | OLISCAG | USGS | SPIReS | MODSCAG | MOD10A1F | MOD10A1F_FSC | VNP10A1F | VNP10A1F_FSC |
|---|---|---|---|---|---|---|---|---|---|---|
| Per-scene median | Bias | 0.004 | -0.004 | 0.003 | -0.001 | -0.001 | -0.071 | 0.028 | -0.095 | -0.027 |
| | RMSE | 0.124 | 0.114 | 0.158 | 0.121 | 0.120 | 0.127 | 0.137 | 0.186 | 0.157 |
| | Precision | 0.994 | 0.992 | 0.963 | 0.938 | 0.942 | 0.962 | 0.962 | 0.894 | 0.894 |
| | Recall | 0.994 | 0.994 | 0.990 | 1.000 | 1.000 | 0.962 | 0.962 | 0.995 | 0.995 |
| | F statistic | 0.991 | 0.991 | 0.973 | 0.961 | 0.956 | 0.944 | 0.944 | 0.931 | 0.931 |
| Per-scene maximum | Bias | 0.082 | 0.107 | 0.170 | 0.176 | 0.191 | 0.065 | 0.234 | 0.055 | 0.186 |
| | RMSE | 0.308 | 0.252 | 0.309 | 0.299 | 0.260 | 0.473 | 0.333 | 0.425 | 0.325 |
| | Precision | 1.000 | 1.000 | 1.000 | 1.000 | 1.000 | 1.000 | 1.000 | 1.000 | 1.000 |
| | Recall | 1.000 | 1.000 | 1.000 | 1.000 | 1.000 | 1.000 | 1.000 | 1.000 | 1.000 |
| | F statistic | 1.000 | 1.000 | 1.000 | 1.000 | 1.000 | 1.000 | 1.000 | 1.000 | 1.000 |
| Per-scene minimum | Bias | -0.089 | -0.119 | -0.086 | -0.214 | -0.159 | -0.445 | -0.252 | -0.383 | -0.216 |
| | RMSE | 0.039 | 0.090 | 0.096 | 0.022 | 0.021 | 0.023 | 0.021 | 0.012 | 0.012 |
| | Precision | 0.770 | 0.704 | 0.762 | 0.158 | 0.300 | 0.333 | 0.333 | 0.000 | 0.000 |
| | Recall | 0.818 | 0.949 | 0.917 | 0.750 | 0.750 | 0.125 | 0.125 | 0.000 | 0.000 |
| | F statistic | 0.829 | 0.815 | 0.832 | 0.261 | 0.429 | 0.182 | 0.182 | 0.000 | 0.000 |
| Basin-wide | Bias | 0.004 | -0.002 | 0.004 | -0.004 | 0.000 | -0.092 | 0.015 | -0.119 | -0.035 |
| | RMSE | 0.047 | 0.061 | 0.062 | 0.067 | 0.065 | 0.125 | 0.085 | 0.150 | 0.083 |
| Validation pixel counts[a] | Total | 326256 | 323843 | 314745 | 19620 | 19622 | 14492 | 14492 | 16600 | 16600 |
| | Median | 8535 | 8553 | 10380 | 163 | 154 | 138 | 138 | 177 | 177 |

[a]Slight differences in pixel counts are possible due to differences in TNs among products and the MOD10/VNP10 data availability (Sect. 4).

**5.2 Product performances under specific conditions**

In addition to the overall statistical assessment, the ability of each product to map snow under specific conditions was assessed. Figures 4, 5, and 6 display the results obtained when the validation datasets were classified by canopy cover fraction, snow cover fraction, satellite view angle, and snow class. Figure 7 displays how error images were generated to highlight the spatial performance patterns at the basin scale. There were a different number of samples in each discretized bin of the four assessment categories, and the data distributions varied among the Landsat 8 products and MODIS/VIIRS products. All snow classes were represented in the MODIS data, with the lowest number of pixels available for the ephemeral snow class. No ephemeral snow class pixels corresponded to snow in the Landsat 8 validation dataset. The VNP101AF product was not available for the dates on which ephemeral snow was mapped by MODIS. The view angle analysis was performed only for MODIS and VIIRS, as Landsat acquires only near-nadir observations. In this study, our sample of snow pixels was representative of all possible MODIS and VIIRS view angles. All snow cover fractions were present in all products. There were a sufficient number of data points available across all snow cover classes to calculate error statistics across the full range of snow cover fractions. Snow cover fractions lower than the minimum snow cover fractions employed by a product (Table 2) were possible because the product and validation datasets were coarsened to account for geolocational uncertainties, and adjacent pixels with no



snow cover could be included in this coarsening procedure. The snow cover distributions obtained for Landsat and MODIS/VIIRS were similar.

The canopy cover distributions differed between the MODIS/VIIRS and Landsat products. The Landsat products contained significant numbers of canopy-free pixels. The relatively coarse spatial resolution daily products from MODIS and VIIRS blended canopy-free with canopy-covered areas, thus reducing the number of canopy-free pixels relative to those in the Landsat products. Very few or no data points were available for the highest possible canopy cover fractions. In the MODIS/VIIRS dataset, only three pixels corresponded to canopy cover fractions above

0.7, so the subsequent analysis was constrained to canopy cover fractions of 0.7 and lower. For the Landsat analysis, only 380 pixels exhibited canopy cover fractions above 0.7. These pixels were used to calculate the F statistic for the Landsat product under high canopy cover conditions; however, due to the poor detection ability (Fig. 4(e)), the bias and RMSE measures were not reliable nor were they representative of the snow fraction mapping capability of this product under such high canopy cover conditions. The bias and RMSE statistics were constricted to pixels with canopy

cover fractions below 0.65 where products can reliably detect per-pixel snow fractions.

       Because the pixel distributions were not consistent across all of the analyzed categories, the statistical results obtained in this section were not directly comparable to the overall results presented in Table 3. Most pixels were found to be highly snow covered, with minimal forest canopies; thus, this pixel type represented a relatively large fraction in the validation dataset applied in the overall validation step, the results of which are listed in Table 3. In this

section, we aimed to understand how well the assessed algorithms perform in specific situations that may not be representative of their overall performances but are still important for understanding their capabilities and limitations.

       Figure 4 displays the snow mapping performances of all Landsat 8 products, showing similar trends associated with the canopy cover and snow cover fractions. All products obtained relatively high RMSE values under dense canopy cover conditions. The OLISCAG and SPIReS Landsat products exhibited similar RMSE trends

associated with canopy cover fractions up to 0.5, while the USGS product showed relatively high RMSEs at low canopy cover fractions (Fig. 4(a)). The USGS product showed positive biases across all canopy cover conditions (Fig. 4(b)). OLISCAG and SPIReS Landsat were relatively unbiased and maintained zero biases under denser canopy cover conditions than USGS fSCA but also exhibited positive and negative biases, respectively, at the highest canopy cover fractions (Fig. 4(c)). We found that no method utilizing Landsat data could reliably map snow cover at the highest

canopy cover fractions available in this study; this finding was clearly shown by the F statistic decreasing to near zero as the canopy cover increased to 0.7 (Fig. 4(e)).

       The products all exhibited relatively high RMSE values at the lowest snow cover fractions (Fig. 4(b)). All three products showed similar declining RMSE trends at high snow cover fractions, but the USGS product had higher RMSE values by approximately 5-15% at low snow cover fractions. All three products exhibited similar but offset

bias trends as the snow cover fraction increased; however, the USGS product had relatively high biases at fSCA values from 0 to 0.6 (Fig. 4(d)). All products overestimated the snow cover fractions corresponding to pixels with relatively low snow cover fractions and underestimated the snow cover fractions corresponding to the pixels with the highest snow cover fractions.

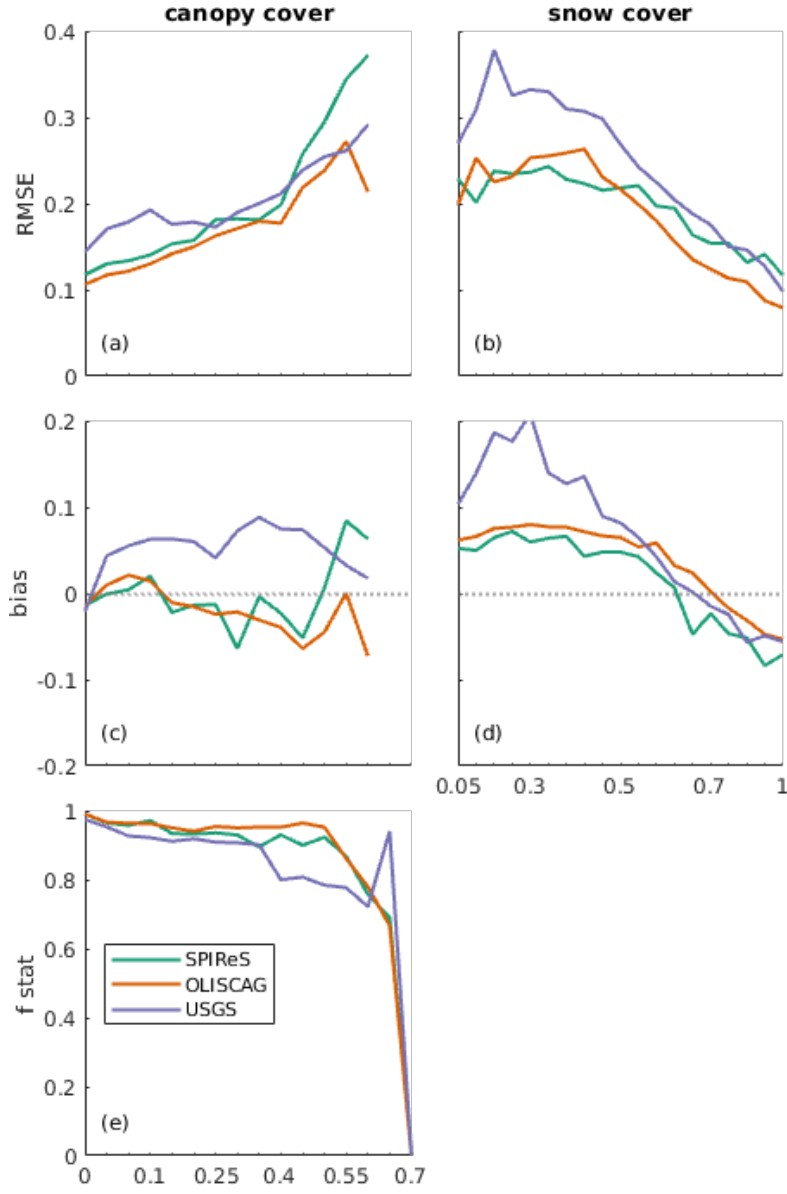

**Figure 4. Snow cover mapping statistics: RMSE (a, b), bias (c, d), and F statistic (e) values calculated for pixels binned by canopy cover fraction (a, c, e) and by ASO-derived snow cover fraction (b, d) from Landsat 8 fractional snow cover products (SPIReS, USGS, and OLISCAG). No F statistic graph is displayed for the snow-covered area due to the low F statistic variability associated with the snow cover fraction.**

More variability was observed between the MODIS and VIIRS algorithm performances (Fig. 5) than among the Landsat 8 products. In the canopy cover analysis (Fig. 5(a)), SPIReS had the lowest RMSE when no canopy cover was present and at canopy cover fractions up to approximately 0.2, with similar errors observed across the 0.2 to 0.45 range. STC-MODSCAG exhibited a similar but slightly worse performance up to a canopy cover fraction of 0.35 but





performed slightly better at fractions from 0.35 to 0.55, when both SPIReS and VNP10 produced lower RMSE values. SPIReS and STC-MODSCAG showed the lowest biases among the analyzed canopy cover range, while the standard
MODIS and VIIRS products exhibited consistent negative biases of approximately -10% under all but the densest canopy cover conditions (Fig. 5(d)). The snow detection ability was poor at the highest canopy cover fractions (i.e., >0.65), so the RMSE and bias results were less reliable and are not shown. The VNP10 snow detection performance dropped off the most under dense canopy cover conditions, with MOD10, SPIReS, and STC-MODSCAG all showing similar performances as the canopy cover fraction increased (Fig. 5(g)).

525       The satellite view angle analysis results showed that the RMSE values of the snow fraction estimates were consistent among all products (except VNP10A1F) and among all view angles. For VNP10A1F, the high RMSE values derived at high viewing angles were significantly reduced when the Collection 5 linear correction was applied (Fig. 5(b)). The Collection 6 MOD10 and Collection 1 VNP10 products showed consistent negative biases across all viewing angles, but these biases were greatly reduced with the implementation of the Collection 5 linear correction
step (Fig. 5(e)). The snow detection ability (i.e., F statistic) was generally unaffected by the satellite viewing angle (Fig. 5(h)).

       The statistical analysis examining algorithm performances across the range of pixel-scale snow-cover fractions showed that, overall, STC-MODSCAG and SPIReS had lower RMSE values and less bias than the standard products (MOD10 and VNP10) (Fig. 5(c), (f)). At the highest snow cover fractions, the RMSE values obtained for the
VNP10 and MOD10 products were high, in the 0.15–0.30 range, and the negative biases were large, in the -0.15 to -0.25 range (Fig. 5(c), (f)).

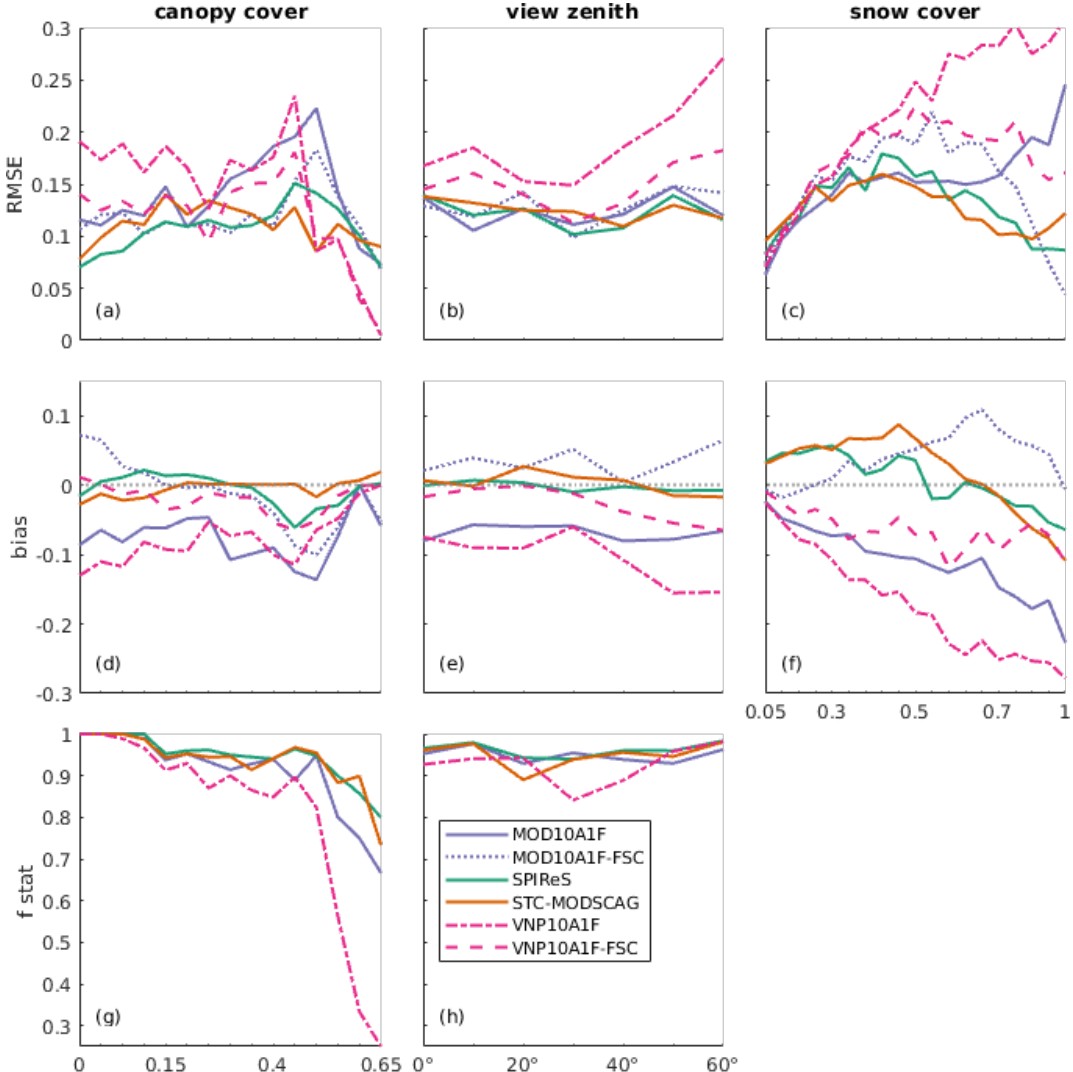

**Figure 5. Snow cover mapping statistics: RMSE (a, b, c), bias (d, e, f), and F statistic (g, h) values calculated for pixels binned by canopy cover fraction (a, d, g), view angle (b, e, h), and snow cover fraction (c, f) using the MODIS and VIIRS snow cover products. The standard products (MOD10 and VNP10) are colored so that each standard product is shown as a solid line, while the fSCA correction is shown in the same color with a dashed line (the F statistic is unchanged by the correction process). No F statistic graph is displayed for the snow cover fraction due to the low variability in the F statistic induced by the snow cover fraction.**

All algorithms were compared to each other in terms of their performances using the global snow classification scheme shown in Fig. 6. Maritime snow corresponded to the largest RMSE values across all products, whereas boreal forest snow had the lowest RMSE values overall, apart from ephemeral snow regions where data were not available for all products. SPIReS Landsat, SPIReS MODIS, OLISCAG, and STC-MODSCAG consistently produced small biases across all snow types, with OLISCAG slightly edging out SPIReS Landsat while the performances of SPIReS MODIS and STC-MODSCAG were virtually identical. The snow detection ability (i.e., F



statistic) was high (>0.98) among all products for tundra, boreal forest, and prairie snow, but the products performed worse for maritime, ephemeral, and montane forest snow. The spectral mixture methods outperformed the NDSI-based methods in these challenging areas.

|  | RMSE | | | | | | bias | | | | | | f stat | | | | | |
|---|---|---|---|---|---|---|---|---|---|---|---|---|---|---|---|---|---|---|
|  | tundra | boreal forest | maritime | ephemeral | prairie | montane forest | tundra | boreal forest | maritime | ephemeral | prairie | montane forest | tundra | boreal forest | maritime | ephemeral | prairie | montane forest |
| USGS | 0.142 | 0.118 | 0.214 | | 0.168 | 0.180 | -0.057 | 0.016 | 0.046 | | 0.017 | 0.063 | 0.999 | 1.000 | 0.918 | | 0.978 | 0.937 |
| OLISCAG | 0.105 | 0.086 | 0.140 | | 0.111 | 0.148 | -0.027 | 0.000 | 0.000 | | 0.000 | -0.031 | 0.999 | 0.998 | 0.945 | | 0.987 | 0.981 |
| L8 SPIReS | 0.124 | 0.069 | 0.160 | | 0.120 | 0.162 | -0.025 | -0.020 | 0.015 | | 0.000 | -0.031 | 0.999 | 0.999 | 0.944 | | 0.988 | 0.953 |
| MOD10 | 0.127 | 0.184 | 0.126 | 0.028 | 0.115 | 0.095 | -0.088 | -0.093 | -0.057 | 0.000 | -0.073 | -0.045 | 1.000 | 1.000 | 0.882 | 0.667 | 0.996 | 0.932 |
| MOD10-FSC | 0.126 | 0.061 | 0.135 | 0.049 | 0.128 | 0.113 | 0.060 | 0.000 | 0.000 | 0.000 | 0.053 | 0.000 | 1.000 | 1.000 | 0.882 | 0.667 | 0.996 | 0.932 |
| VNP10 | 0.230 | 0.202 | 0.190 | | 0.166 | 0.130 | -0.166 | -0.158 | -0.103 | | -0.107 | -0.059 | 1.000 | 1.000 | 0.850 | | 0.989 | 0.922 |
| VNP10-FSC | 0.167 | 0.081 | 0.162 | | 0.130 | 0.122 | -0.051 | -0.044 | -0.033 | | 0.000 | 0.000 | 1.000 | 1.000 | 0.850 | | 0.989 | 0.922 |
| STC-MODSCAG | 0.097 | 0.091 | 0.128 | 0.053 | 0.113 | 0.119 | -0.017 | 0.000 | 0.000 | 0.000 | 0.000 | 0.000 | 1.000 | 1.000 | 0.925 | 0.667 | 1.000 | 0.969 |
| MODIS SPIReS | 0.105 | 0.107 | 0.123 | 0.048 | 0.089 | 0.106 | -0.016 | 0.000 | 0.000 | 0.010 | 0.000 | 0.000 | 1.000 | 1.000 | 0.927 | 0.667 | 1.000 | 0.965 |

**Figure 6. Heatmaps of the algorithm performance results (colored by RMSE, bias, and F statistic) corresponding to seasonal**
**snow cover types. The Landsat algorithms are denoted in the first three rows. The black boxes indicate instances of insufficient validation data.**

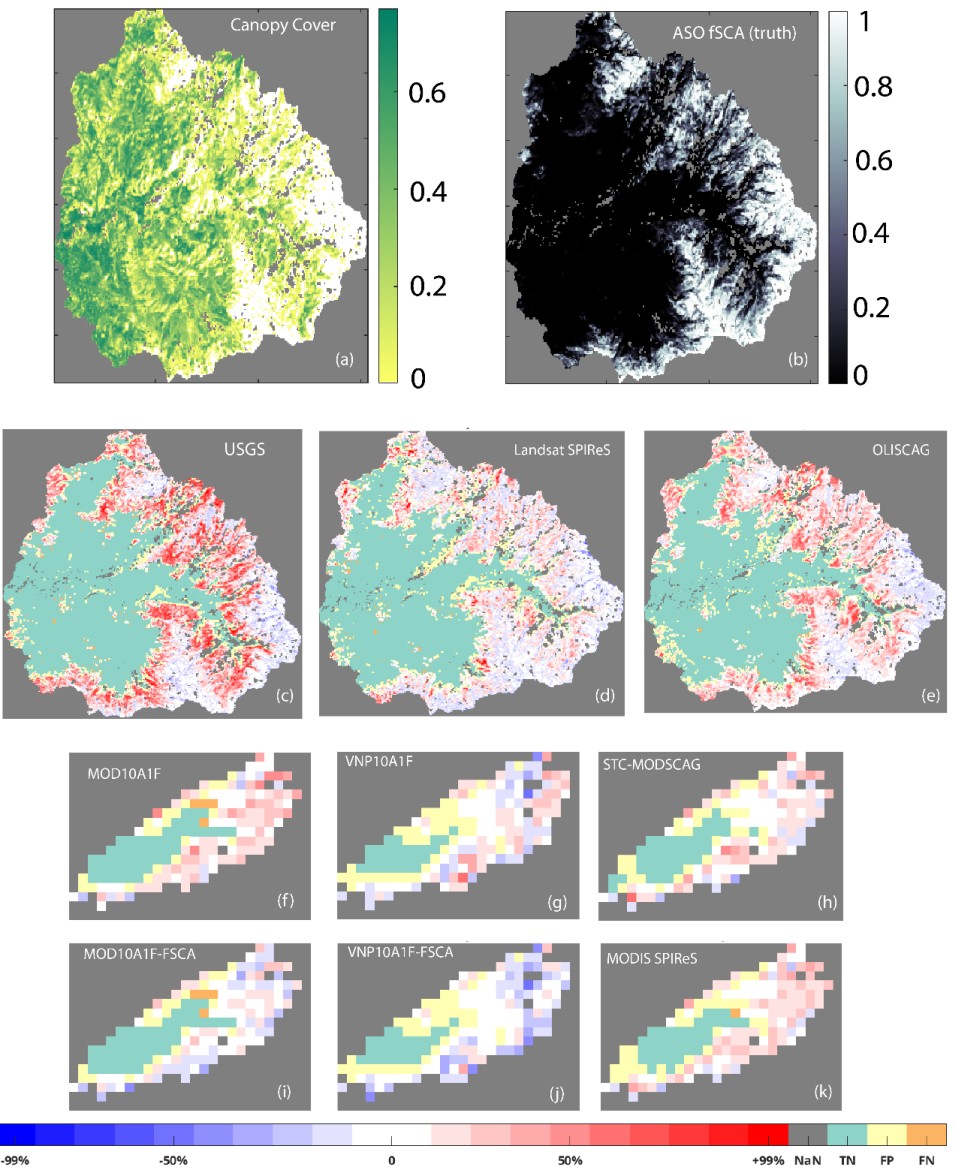

**Figure 7 Merced River watershed, Sierra Nevada, California, on 07 May 2020; Yosemite Valley is approximately in the middle of these images displaying the 835 km² watershed. Subpanel (a) shows the canopy cover fraction. Subpanel (b) shows the fSCA information obtained from ASO at a 120 m pixel resolution. Subpanels (c-e) show comparisons of the Landsat products against the reference snow cover data from ASO at a 120 m pixel resolution. Subpanels (f-k) display comparisons of the MODIS/VIIRS products at a 2 km pixel resolution. The red, white, and blue pixels indicate locations where a product accurately detected snow cover (TP) but, compared to the ASO truth data, was found to have generated a positively biased result (red), an unbiased result (white) or a negatively biased result (blue). The orange pixels denote FNs (omission errors) in which the product did not correctly detect snow in the indicated location, while yellow pixels indicate FPs (commission errors) in which the algorithm incorrectly identified the presence of snow cover compared to the ASO truth dataset. The gray pixels (not-a-number (NaN) cells, outside the Merced basin) and green pixels (TNs) were not included in the analysis or used in any of the error statistic calculations performed in this paper. Note that the spatial projections differ (UTM for**



**Landsat 8 vs. sinusoidal for MODIS and VIIRS) among the products in the three panels, but all panels display the same area.**


## 6 Discussion

NDSI snow cover mapping algorithms and spectral mixture analysis models designed for use with Landsat, MODIS, and VIIRS products were validated using snow cover maps derived from all available ASO snow depth measurements collected in California and Colorado over water years 2013-2020. The spatial extent of these ASO validation data

enabled the snow mapping algorithms to be validated over a wide range of conditions. Our insights extended past the overall algorithm performances to explore how well the analyzed products perform in specific situations in which accurate snow cover mapping is important but systematic biases or errors may emerge; these situations included considerations of the canopy cover conditions, per-pixel fractional snow cover, satellite view angle, and global snow type classification. In addition, we qualitatively assessed the overall spatial snow cover patterns at the river-basin

scale.

Landsat data have traditionally been used in spatial validations of MODIS snow products (Rittger et al., 2013; Hall and Riggs, 2007) or for calibration tasks (Salomonson and Appel, 2004, 2006). In this study, we showed that while Landsat data have generally high snow detection and fSCA mapping capabilities, they are imperfect and do not show the same patterns as the coarser-spatial-resolution MODIS and VIIRS products. The RMSE and bias values

decreased as the snow cover fraction increased for all Landsat products. The RMSE values increased and the F statistics decreased as the canopy cover density increased. Thus, for these important pixel categories, Landsat products are an imperfect measure of the fractional snow cover on the ground, while aerial lidar-based approaches can deliver improved satellite product validations.

### 6.1 Comparison between standard products and spectral mixing methods

Many users rely on standard snow cover products. For Landsat 8, the USGS Collection 1 canopy-corrected fractional snow cover product exhibited a slightly worse performance than OLISCAG and SPIReS across snow cover types (Fig. 6) and across ranges in canopy cover and snow cover (Fig. 4). The largest difference between this standard product and OLISCAG or SPIReS was the bias metric, as neither OLISCAG nor SPIReS produced significant overall biases when mapping the snow cover fraction. The USGS Collection 1 product is available only as a canopy-corrected

version over the western US and Alaska. However, the USGS Collection 2 product expands to include a viewable snow cover product and covers the northern portion of the US and the Aleutian Islands. Both OLISCAG and SPIReS can be run for any region globally. While OLISCAG will be released publicly in 2024 pending a processing transfer to the NSIDC DAAC, the SPIReS algorithm is currently open-source (https://github.com/edwardbair/SPIReS) and a viable alternative for creating high-quality snow cover fraction maps globally.

While STC-MODSCAG is produced operationally for the Western US and the Indus River basin (Rittger and Raleigh, 2022), the standard NASA global MODIS and VIIRS snow products (MOD10 and VPN10) no longer output fractional snow cover data, as NDSI values are produced instead. The results of this study suggest that the standard (NDSI-based) MODIS and VIIRS products should not be interpreted as fractional snow cover estimates. We



found the now-decommissioned Collection 5 linear correction method to be effective for improving the quality of
these standard products across all metrics for which they were assessed, though spectral mixture approaches performed
better in every metric assessed. The results obtained in this study support the application of the Collection 5 conversion
of NDSI to fSCA (Eq. (3)) in the Collection-6 standard snow products derived from MODIS or VIIRS. This conversion
method does not seem to consistently cause large errors, even when applied to VNP101AF, for which this correction
was not designed. A VIIRS-specific correction method may yield further improvements to the VIIRS products, as
VNP10A1F performed worst across all metrics assessed in this study.

The poor performance of the VIIRS product may have stemmed from the different bandpasses used by VIIRS
to calculate NDSI compared to the MODIS-utilized bandpasses (Fig. 2). The VIIRS visible band is centered on
relatively long wavelengths compared to the MODIS visible band, where snow reflectance is impacted by both the
fraction of the pixel covered by snow as well as the grain size of the snow, thus supporting simultaneous solutions for
the snow fraction and grain size, as is performed with spectral mixture approaches. A significant limitation to NDSI
methods is their need to be recalibrated to achieve relative accuracy when new sensors are introduced. The VIIRS
recall and precision metrics, which were unaffected by the linear correction applied to the snow cover fraction, were
worse than the corresponding SPIReS MODIS and STC-MODSCAG values. Newer spectral mixture analysis methods
consistently performed as well as or better than the standard MODIS and Landsat products across all measures
considered in this study. Spectral unmixing approaches have already been shown to be robust when transitioning
between sensors like MODIS and VIIRS (Rittger et al., 2021a). Based on this work, we expect SPIReS to have similar
performances on VIIRS and expect these spectral unmixing algorithms to be insensitive to bandpass differences among
other sensors such as Sentinel 2a/2b (Bair et al., 2022) and the upcoming Thermal infraRed Imaging Satellite for High-
resolution Natural resource Assessment (TRISHNA) mission. Global standard MODSCAG and VIIRSCAG products
are currently being undertaken by the NSIDC DAAC, and SPIReS MODIS will be produced operationally for North
America, Greenland, and High-Mountain Asia as part of Snow Today at NSIDC (Rittger and Raleigh, 2022).

## 6.2 Snow class insights

Mountainous regions contain some of the most diverse assemblages of snow classes among any biome around the
world. In this work, we assumed that the snow class mapping performance of each product exhibited in our validation
scenes was similar to the expected performance of the corresponding algorithm when assessing the same snow classes
in different geographical regions. This logic enables us to use this validation method to assess algorithm performances
in other global regions where high-quality, high-spatial-resolution validation data (e.g., airborne lidar-derived snow
depths) are not available.

Across all snow types, the spectral unmixing approaches SCAG and SPIReS performed as well as or better
than the standard products currently available to global users. This finding gives credence to the assertion that spectral
unmixing is a viable approach for mapping snow globally and that this method can improve our ability to map
numerous seasonal snow cover types at the global scale. In prior decades, the computational requirements associated
with spectral unmixing were seen as a barrier to operationalizing the methods into global products. Recent
improvements in computing power and significant advances in algorithms have reduced the number of calculations





required by 10-100× (Bair et al., 2021a), thus furthering spectral mixture analysis methods to a point where it is no longer a barrier to run these algorithms on global datasets.

For all products assessed in this work, detecting snow (see F statistic results) was most challenging in relatively warm, forested areas (e.g., montane forest and maritime regions). For the Landsat products, the RMSE values were highest for forested snow types. Ephemeral snow and forest snow were the most challenging snow types

to detect (based on the F statistic results). Because of internal fSCA detection thresholds established for some products, it was difficult to map low-fSCA ephemeral snow. The assessed spectral unmixing approaches, apart from STC-MODSCAG, do not map snow at the lowest fSCA values (Table 2) because the probability of obtaining FP detections characterized by bright non-snow surfaces increases substantially at very low (<0.08-0.1) fSCA values. FPs create the need for persistence filters, thus hindering the ability of these algorithms to map ephemeral snow. Ephemeral snow

was also the least-represented snow cover type in our analysis, and additional investigations are needed to better characterize ephemeral snow mapping capabilities involving multispectral satellites. Geostationary satellites such as the Geostationary Operational Environmental Satellite R-series (GOES-R), Himiwari, or Korean Multi-purpose Satellite (KompSAT) provide better temporal sampling for ephemeral snow detection than the once-daily detection abilities of polar orbiting satellites. However, the snow mapping approach utilized by the National Oceanic and

Atmospheric Administration (NOAA), while technically an application of spectral mixture analysis, uses only a single band and a background endmember (Romanov et al., 2003). In forests, we observed significant declines in the performances of all algorithms compared to their snow mapping abilities observed when the satellites had a clear view of the ground from space.

**6.3 Canopy cover insights**

This study augments prior evaluations of satellite-based snow mapping performances under dense canopy cover. STC-MODSCAG was validated under a dense canopy by Rittger et al. (2020) and Raleigh et al. (2013) with ground-based temperature sensors. While those studies considered fewer study sites than were assessed herein, their evaluations were more temporally continuous. In this study, results showed a lower F statistic at the highest canopy cover density than that obtained at the highest-canopy-cover-density site in Rittger et al. (2020). This was probably due to the

temporal distribution of the ASO validation dataset; flights are temporally focused after peak SWE occurs, when snow is more likely to be only on the ground and no longer on top of the canopy as is observed following storm events. SPIReS was also validated by Bair et al. (2021a) using an ASO dataset smaller than that used herein, and the results showed that the F statistic fell to approximately 80% for dense canopies; the dataset utilized in that study contained scenes representing only California.

The three Landsat approaches validated in this study all involve the use of different canopy-correction methods. SPIReS and OLISCAG produced nearly identical RMSE variabilities associated with the canopy cover density, exhibiting relatively low RMSE values at low canopy cover densities and high RMSE values at the highest canopy cover densities compared to those obtained with the USGS Collection 1 fSCA product. SCAG had the most robust canopy cover correction for the Landsat products when ranked based on the minimum snow cover estimate

bias and RMSE across all canopy cover densities. From the example shown in Fig. 7, SPIReS Landsat showed the





lowest snow cover fraction bias under a forest canopy, though Fig. 4 indicates the near-equal performances of SPIReS Landsat and OLISCAG. The USGS product employs a spatial replacement step different from that of SPIReS Landsat (Sect. 4.1.5). This approach is the likely reason behind the higher F statistics observed for the USGS product at relatively high canopy cover densities before its snow detection ability dropped off, similar to the trends observed for

the other products. Above a canopy cover fraction of approximately 0.65, none of the Landsat algorithms reliably mapped snow cover under the forest canopy, though few pixels were available for testing the algorithm performances under very dense canopy cover conditions. At canopy cover fractions ranging from 0.5 to 0.6, viewable snow cover mapping from multispectral optical satellite data becomes challenging if the mapping process is conducted based only on the spectral characteristics of individual pixels. Ancillary spatial or other data are needed to improve snow cover

estimates in pixels containing dense canopy cover. At these extremely high canopy cover densities, it is not only difficult to detect snow; it is also difficult to correct the actual fractional amount of snow present on the ground.

All approaches considered herein estimated more snow cover in pixels with dense canopy cover than the lidar validation dataset. It is unclear if these overestimates were caused by the canopy correction methods inherent to the remote sensing products, an issue associated with no ground returns from the lidar for some dense forest canopy

locations thereby skewing analysis, or both. At the meter scale, airborne lidar with typical point densities will not receive ground returns from all pixels covered by dense forest canopy (Zheng et al., 2016). ASO is also known to not always receive ground returns for all grid cells, with the issue most prevalent for heavily forested locations. Cao et al. (2018) acknowledges that ASO ground point densities decrease non-linearly as canopy cover and vegetation height increases. Across all forested locations analyzed by Currier et al. (2019), 17% of the forested pixels has no ground

returns from ASO. It is likely that some of the positive biases reported herein for snow cover mapping under dense canopies by the remote sensing products are due to FN snow depth returns at the 3 m scale by ASO reducing the ASO fSCA values at the validation resolution.

The USGS L3 fSCA canopy correction method predated recent studies that have shown that forest snow cover is dependent on both the time of year and type of forest. Forest dynamics are complex; different accumulation

and melt patterns can be observed depending on specific forest characteristics (Dickerson-Lange et al., 2021), and variabilities arise in the relationship between forest fSCA values and adjacent open-area fSCA values due to various forest dynamics controlled by the canopy cover density, temperature, and aspect (Safa et al., 2021). Both of these past studies observed sparser late-season snow cover in denser canopy-covered areas. ASO usually performs springtime data acquisitions; at this time, the forest snow cover is more variable than that in mid-winter. Our study highlights and

confirms that fSCA values below 1 are pervasive under forest canopies and that more complex canopy-correction methods are needed if estimates of snow cover on the ground under forest canopies are to be improved. Additionally, snow present on the ground is simply not detectable under canopy cover fractions above 0.65, though snow in the canopy can often result in positive snow detections. Additional spatial data and ancillary snow information may help significantly improve our ability to map snow on the ground under dense forest canopies.

Regarding the MODIS and VIIRS products, due to their canopy-correction methods and ability to integrate surface information from multi-day measurements obtained at various sensor zenith angles, the spectral unmixing approaches showed improved snow detection abilities under high canopy cover fractions compared to the standard





products and the Landsat products. Additionally, the larger pixel sizes in the MODIS and VIIRS products allow canopy gap areas where snow detection for fSCA>0.1 is trivial, to be integrated in many pixels that contain canopy cover. In
the Landsat products, varying bias patterns associated with the canopy cover density were observed under the same approaches; this finding may have been influenced by scale-dependent differences in the canopy cover distribution across the study area. There is a tradeoff between the aggressiveness (i.e., correcting to fSCA=1) of the employed canopy-correction method and the overall bias level observed in the results derived across the entire scene, potentially inducing major spatial issues when estimating snow cover fractions. Raleigh et al. (2013) and Rittger et al. (2020)
previously reported a similar issue, which was particularly problematic when the canopy cover density exceeded the viewable snow cover fraction.

       Full waveform lidar data can be used to parameterize the viewable gap fraction in forests (Liu, 2008; Xin et al., 2012) to set an a priori expectation of the maximum viewable snow for a canopy correction step. While we used only a lidar snow depth product with a spatial resolution of 3 m, as no full waveform dataset was available, good
models exist for understanding what nadir and off-nadir sensors can see while considering a theoretical maximum. With original, full-waveform lidar data, one can also derive the necessary parameters for gap fraction modeling (Morsdorf, 2006; Zhao et al., 2012). SPIReS enables users to make parameterization adjustments for various canopy crown sizes and tree sizes in the fSCA canopy correction step, a procedure not possible with the SCAG or USGS products. Analyses with lidar datasets constructed in ranges like the Olympic Mountains could also shed more light
on snow cover conditions in dense forests.

## 6.4 Snow covered area insights

 All of the spectral unmixing approaches considered herein exhibited significant negative biases at high snow cover fractions. We posit that this finding can be attributed to two issues. First, it is impossible to obtain positive biases for fully snow-covered pixels, as snow cover cannot exceed fSCA=1. Second, it is difficult to find a spectral unmixing
solution for fSCA=1 that is substantially better than the solutions available for slightly lower snow cover fractions when dealing with many pixels. This is especially true when shading is present in the pixels, as shading has been shown to significantly lower the apparent albedo and snow cover reflectance (Bair et al., 2022).

       The combination of systematic snow cover overestimates in forested areas with systematic underestimates of alpine snow cover led to the overall bias estimates being close to zero for all spectral unmixing products. The spectral
mixture analysis methods, when applied to the Landsat and MODIS products, exhibited the best performance by achieving snow cover products with low bias and with the best results across the assessed snow cover and canopy cover ranges. Error images, like those shown in Fig. 7, enabled us to identify systematic errors in snow cover fraction estimates and update the models, thus supporting a continuous development approach to snow mapping. For example, we discovered that the opposing highly biased estimates derived in pixels with dense canopy cover and high fractional
snow cover conditions can offset to generate an overall low bias. Figure 7 shows a spatial pattern in which underestimates (blue) are generally located in canopy-free areas and overestimates (red) are generally located in canopy-covered areas. The Landsat products consistently overestimated the snow cover fraction under dense canopy cover conditions and incorrectly detected snow around the snowpack boundaries near the lowest snow cover fractions.



A possible explanation for this finding is that the interactions among the ground instantaneous field of view (GIFOV),
point spread function, and pixel size led to pixel overlaps in these regions and to snow being identified within the
GIFOV but outside the final pixel ground sample distance, thus causing FP detections and snow fraction overestimates;
alternatively, the ASO snow detection map may be negatively biased.

### 6.5 View angle insights

A surprising result obtained from this study is the insensitivity of the snow mapping algorithms to the view angle of
the MODIS or VIIRS sensor given previous findings of MODSCAG view angle sensitivities with respect to forests
(Rittger et al., 2020). However, Rittger et al. (2020) investigated raw or initial retrievals from MODSCAG, while both
STC-MODSCAG and SPIReS MODIS incorporate pixel-based weighting schemes designed in consideration of the
sensor view angle (Dozier et al., 2008). The VNP10 product showed some increasing RMSE values as the view angle
increased, but this trend was not monotonic, and the lowest RMSE values were observed in the view angle range of
~20–30°. However, both the corrected and uncorrected VNP10 and MOD10 results exhibited relatively large biases
at the largest view angles compared to the STC-MODSCAG and SPIReS results. We expected to see a stronger
relationship between the view angle and errors in the standard products than in the products constructed using spectral
unmixing algorithms. A possible explanation is that the band ratio method experiences reduced view angle impacts
because 1) both the VIS and SWIR observations are affected and 2) dividing the VIS/SWIR reflectance difference by
the sum of the VIS/SWIR reflectance (originally designed as a way to account for atmospheric differences when NDSI
is applied to derived top-of-atmosphere reflectance) compensates for the view angle-related errors.

Both STC-MODSCAG and SPIReS include temporal weighting and smoothing routines that weight data
collected at nadir view angles higher than data collected at off-nadir view angles. The fact that the view angle was not
found to be strongly or consistently related to the error statistics indicates that these weighting schemes are effective.
The coarsening of the MODIS and VIIRS products to the 2 km scale for use in our evaluations may have also
minimized the view angle effect by increasing the pixel size.

### 6.6 Landsat vs. MODIS insights

In the study area, the Landsat-derived data contained significantly more canopy-free pixels than canopy-covered pixels
compared to the coarser-resolution MODIS and VIIRS products (Fig. 4(d)). The 30 m scale observations lend credence
to the value of techniques that can leverage canopy-free observations corresponding to locations at which MODIS
cannot obtain canopy-free observations, thus improving our ability to detect various snow properties from space
(Rittger et al., 2021b). However, as the canopy cover increases, all available algorithms face challenges when mapping
snow; the detection abilities of these algorithms deteriorate rapidly with increasing canopy cover conditions. The F
statistic values decreased with increasing canopy cover densities for all Landsat products (Fig. 5(g)), implying that
Landsat cannot identify all snow in forested landscapes and is thus an imperfect validation measure for coarser-
resolution products. The ability of lidar instruments to both penetrate forest canopies and measure snow at sufficiently
fine spatial scales to eliminate the need for fSCA and instead produce binary snow maps provide lidar-based methods
with the improved ability to validate other snow products, such as those derived from multispectral sensors in forested





areas. Prior studies have also shown that high-spatial-resolution commercial satellite data with spatial scales ranging
from 0.5-4.0 m, the same spatial scales as those of lidar data, can be used to reliably map snow cover fractions for use
in validations performed in canopy-free areas and to reliably detect snow cover presence under canopies; however,
high-spatial-resolution commercial satellite data cannot be used to measure snow cover fractions under moderate
canopy cover conditions (Bair et al., 2016; Bair et al., 2021a). Airborne lidar data and spaceborne high-resolution
commercial data provide alternative validation solutions to using standard multispectral satellite data (e.g., Landsat
and Sentinel-2 data).

## 7 Conclusion

The process by which snow cover is mapped using data collected by multispectral satellites has matured through the
development of a variety of methods. Snow cover varies at a finer spatial resolution than that captured by multispectral
sensors; thus, understanding how well various methods work across a range of snow cover fractions and in a variety
of landscapes is critically important for recognizing global snow mapping capabilities. In this study, we examined
how well various products perform across various canopy cover densities, per-pixel fractional snow cover conditions,
satellite view angles, and global snow type classifications. Supporting previous work, we found that spectral unmixing
algorithms perform better than standard NDSI-based products. Aerial lidar retrievals indicated that this finding holds
true across a diverse range of snow and forest cover conditions and among all global seasonal snow cover
classifications. Spectral unmixing methods have reached a level of readiness that allows them to be deployed with
modern computing techniques to advance snow cover mapping at the global scale.

## 8 Data and code availability

All Airborne lidar datasets were obtained from NSIDC (https://nsidc.org/data/aso) and ASO, inc
(https://data.airbornesnowobservatories.com/). The key tables which match individual ASO lidar flights to the MODIS
and VIIRS tiles, Landsat path/rows, and Landsat ARD tiles are available as a supplemental zip file to this manuscript.
The Canopy Cover data used in the validation is publicly available at https://www.mrlc.gov/data. Sensor view angle
data came from publicly available MOD09GA and VNP09GA datasets from NASA DAACs. VNP10A1F and
MOD10A1F datasets were downloaded from NSIDC in May 2022 (https://nsidc.org/data/). SPIReS datasets used in
this analysis are hosted online. MODIS SPIReS (https://snow.ucsb.edu/products/SPIRES/WUS) and Landsat SPIReS
(https://snow.ucsb.edu/products/SPIRES/Landsat8/TCD_fSCA_validation_2022). The SPIReS codebase is publicly
available on GitHub with a tag for the version used in this validation analysis
(https://github.com/edwardbair/SPIRES/releases/tag/v1.1). STC-MODSCAG and OLISCAG data from this paper is
available (ftp://snowserver.colorado.edu/pub/fromRittger/20220801_TCS_Validation_of_SCAG) and will be
permanently stored on Zenodo with a DOI if this paper is accepted. The SCAG code is closed source and unavailable.



## 9 Conflicts of interest

A co-author is a member of the editorial board of The Cryosphere. The peer-review process was guided by an independent editor, and the authors have no other competing interests to declare.

## 10 Acknowledgements

This material is based upon work supported under the US Army Research, Development, Test and Evaluation Program, Program Element 0602144A, under the authority of the Broad Agency Announcement Program and the Cold Regions Research and Engineering Laboratory (ERDC-CRREL), Contract No. W913E521C0001. M. Raleigh was supported in part by funding from NASA under grants 80NSSC22K0685 and 80NSSC22K0928. K. Rittger was supported by funding from NASA under grants 80NSSC18K0427 and supplements to support Snow Today at NSIDC and 80NSSC20K1721 as well as NOAA NA18OAR4590367. Timbo Stillinger and Edward Bair were supported by NASA awards 80NSSC21K0997, 80NSSC20K1722, 80NSSC20K1349, 80NSSC18K1489, and 80NSSC21K0620. Mary Joe Brodzik processed the MODSCAG data used in this paper.

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
