# Peer review of "Landsat, MODIS, and VIIRS snow cover mapping algorithm performance as validated by airborne lidar datasets"

_The Cryosphere, 2022_

## Author Comment (AC1)

Authors response to RC-1

The reviewer's comments are in Cambria font.

The authors' responses are in blue Calibri font and are indented

The paper provides a detailed overview of the quality of standard snow products.

While the used analysis methods are not new, the absolutely new approach lies in the fact that it is the only work, I know so far, which comprehensively evaluated NASA snow cover products with precision lidar data. It gives an excellent overview on the quality of the products. This work is very useful for all who are working in the field of snow remote sensing.

The minor corrections that have to be done are only technical ones:

- Line 207: wrong figure given. It is fig. 2 not fig. 1

    Thank you for catching this error. Fixed.

- Figures 4 and 5: Labelling of the x-axes; it would be more convenient, if each of the graphs would show a labelling of the x-axes plus name of the axe (% of snow cover, % of canopy cover)

    An x-axis label has been added to figure 4 and 5. Because the x-axis is the same for all rows in each column in each figure, we have only placed a single x-axis that is labeled at the bottom of each column.

---

## Author Comment (AC2)

**Authors response to CC-1**

The reviewer's comments are in Cambria font.

**The authors' responses are in blue Calibri font and are indented**

It is interesting work on fractional snow cover estimation using multisource remote sensing data, including Landsat, MODIS, and VIIRS. Authors evaluate various algorithm and fractional snow cover products.

I have some comments for this excellent work:

1): Line 39-40: Please add a reference that can demonstrate the statement of "... the fraction of precipitation that falls as rain, rather than as snow ..."

Two references have been added to further support this statement. https://doi.org/10.1029/2011GL046976 https://doi.org/10.1029/2007JD008397

2): Line 110: "but these approaches are not available for the dates or areas considered in this analysis". I don't understand this sentence's meaning, can you give more explanations why these two methods (MODiMlab and SnowFrac) were not used in this study.

Neither of these methods are in production and to the best of our knowledge they were never run for the study area for the dates of the airborne lidar validation datasets. The goal of this study was to assess currently available operational snow cover mapping algorithms and state of the art spectral unmixing methods. These two legacy methods were included in the introduction to give a greater context to the history of snow mapping products and past validation efforts.

3): Lines 841-843: Changed "Bair, E. H., Stillinger, T., and Dozier, J.: Snow Property Inversion From Remote Sensing (SPIReS): A Generalized Multispectral Unmixing Approach With Examples From MODIS and Landsat 8 OLI, Ieee T Geosci Remote, 59, 7270-7284, 10.1109/Tgrs.2020.3040328, 2021a" to "Bair, E. H., Stillinger, T., Dozier, J., Snow Property Inversion from Remote Sensing (SPIReS): A generalized multispectral unmixing approach with examples from MODIS and Landsat 8 OLI, IEEE Trans. Geosci Remote Sens, 59, 7270-7284, 10.1109/TGRS.2020.3040328, 2021a"

Thank you we have updated reference to correct format in final publication.

4): Lines 847-848: Please modify the citation stye of "Bair, E. H., Stillinger, T., Rittger, K., and Skiles, S. M.: COVID-19 Lockdowns Show Reduced Pollution on Snow and Ice in the Indus River Basin., P Natl Acad Sci USA, 118, 2021b."

Thank you we have updated reference to correct format in final publication.

5): Line 150: What's your meaning of "fSCA can depend on the snow climate"? Please give more explanations. Do you would like say that the fSCA depend on snow depth, density, and grain size?

This statement is aligned with the two papers cited in the sentence (Liston, 2004, Clark et al, 2011). We are providing reasoning from the broader community on why it is important to understand how snow cover mapping performs in various snow climates. Snow cover evolution

and the spatial patterns of snow cover are partially driven by the unique energy and moisture fluxes that occur in various snow climates. This is a reason that drove us to validate fSCA by snow climate in this paper. (See Figure 3, Figure 6, and Section 6.2)

**6): Line 317: Please explain the selection of a threshold of 0.01 for converting fSCA to binary snow. Why not 0.1, or 0.2, 0.3?**

Line 319 in the original submission gives the explanation for the threshold choice. This threshold was not an fSCA threshold, but instead a data cleaning step during spatial coarsening. A threshold of 1% (0.01) was chosen to remove artifacts from upscaling the 3m snow depths to the final validation resolution. Depending on the geospatial location of a 3m snow depth measurement relative to the final validation resolution pixel, incredibly small snow fractions are possible as only a small portion of a single 3m snow depth pixel might overlap the final validation pixel. (ASO fSCA<<<0.01) . The very low threshold of 1% fSCA, below the detection limit for snow cover from these sensors, was chosen to eliminate these values from the validation dataset.

**Section 4.3.1 Upscaling**

7) What's the difference of the validation experiment between that was in the data original scales (463 m, or 373m, or 30 m) and that was in the upscaling scales?

We did not perform any validation at the original data scales due to uncertainty in the geolocation of individual multispectral satellite pixels. Prior validation studies cited in our paper have taken this same upscaling approach to validation to ensure that there is overlap between the validation data and satellite observations.

8) Fig. 5: When the canopy cover is over 0.5, these six products have lower RMSEs (Fig. 5a), however, f test are decreasing so fast (Fig. 5g). Why? It is so abnormal. Compared to Fig. 4, low f-test is corresponding to higher RMSE.

In addition, there is higher RMSE for VNP10A1F data at view zenith angle > 50 ° conditions (Fig. 5b), however, its f-test is so high, closing to 1 (Fig. 5h). Please confirm your data.

Thank you for pointing this out. This is an interested case of trying to validate with a small sample size. The F stat is a measure of detection of pixels with snow cover (a Binary detection measure). At high canopy cover, none of the products are great at detection of snow covered pixels, with VNP10A1F showing the worst performance.

Bias and RMSE calculations do not account for false negatives. There are numerous false negatives at high canopy cover values, as seen in the low F stat values. The RMSE and bias are only calculated from the set of true positive detections of snow cover.

Our probability of detecting a snow covered pixel is low for high canopy covered areas, but when we do detect snow, we do a good job of estimating the snow cover for that pixel.

The low RMSE and bias have to do with the combination of low sample size and the aggressive canopy cover adjustments for high canopy cover pixels, that tend to guess snow fraction correctly. The paragraph that starts at line 475 in the initial submission details the very low sample size at these higher canopy cover fractions. The reader is caveated to understand these

sample sizes for the highest canopy cover fractions. For the Landsat data, there was a larger sample size at the high canopy cover fractions as so the sample size enables a better estimate of RMSE and bias.

9) Figs. 4 and 5: The label "snow cover" in these two images are so confusing. I suggest that you modified it to another label word.

Additional clarification has been added to the figure caption.

---

## Author Comment (AC3)

The reviewer's comments are in Cambria font.

> The authors' responses are in blue Calibri font and are indented

General comment

Overall this is a well-constructed paper that offers a new development in the assessment of fSCA products delivered by several spaceborne sensors (MODIS, VIIRS, Landsat) against snow cover as determined from lidar-derived snow depth maps provided by the Airborne Snow Observatory (ASO).

I found the discussion particularly well laid out and containing many useful points that will serve the community well, particularly as we consider more MODIS – VIIRS data continuity for snow cover mapping.

I recommend publication once the specific comments below have been addressed. I have also noted technical/editorial points below that should be addressed.

> Thank you for your thorough and thoughtful review of the manuscript.

Specific comments

The authors frequently refer to MODIS as having 463 m resolution, which is a fair point to make, but I think this should be better supported by relevant references/explanation to ensure that readers understand the reason for the distinction. This is particularly the case when most documentation (including that cited by the authors) refers to 500 m resolution, and where most readers would be interacting primarily with MODIS products that are resampled to a 500 m grid.

> Clarification has been added to line 62 of the paper. The sentence now reads:

> *"At a 463 m spatial resolution (i.e., the sinusoidal tile product grid cell size for Moderate-resolution Imaging Spectroradiometer (MODIS) data with Ground Sample Distances of 500m at nadir) (Campagnolo and Montaño 2014)"*

> > This is an important technical detail for the appropriate use and validation MODIS data. All of the MODIS data docs (e.g. Users Guides) state 250m/500m/1km as "grid cell" size, but those are Ground Sample Distances (GSD) at nadir, not the grid cell sizes.

> > MODIS uses an authalic sphere (no flattening) of r=6371007.181 m radius & there are h=36 horizontal tiles covering all longitudes. Thus, each tile's width is w=2*pi*r/h = 1,111,950.5 m. For the three GSDs mentioned above, the grids are 4800, 2400, and 1200 pixels wide. Thus, the cell size for each is 231.7, 463.3, & 926.6 m.

> > Section II in the following paper confirms the pixel grid size is 463 and not the same as the 500m GSD.

M. L. Campagnolo and E. L. Montaño, "Estimation of Effective Resolution for Daily MODIS Gridded Surface Reflectance Products," in IEEE Transactions on Geoscience and Remote Sensing, vol. 52, no. 9, pp. 5622-5632, Sept. 2014, doi: 10.1109/TGRS.2013.2291496.

A central challenge to this approach is the conversion of snow depth (in this case from the ASO maps) to snow cover, in order to compare against fSCA products. Here, the authors used an 8 cm threshold to convert snow depth to snow cover in the ASO data, based on reported MAE of 8 cm for ASO snow depth products (Painter et al., 2016). I found the justification for this somewhat lacking, noting that Painter et al. state that (as of 2016), ASO data had not been subject to a full accuracy assessment in forested areas and steep terrain, and the 8 cm MAE is determined with respect to manual snow depth measurements – a comparison which can itself be problematic. Given the role of the ASO snow depth products as the reference dataset underpinning the analysis presented in this manuscript, I think there is scope for a more robust approach to converting ASO snow depth to snow cover. This could include, for example, analyzing residuals in ASO products for snow-free areas, and comparing ASO snow depth maps with optical imagery acquired contemporaneously (if/when available). It would be interesting to consider more fully the quality of ASO snow depth maps in steeper terrain where, for example, even relatively small co-registration errors may result in large errors (both positive and negative) in snow depth estimation. Furthermore, discarding snow depths <8 cm seems a bit blunt given that depths of much less than that can certainly contribute to a snow signal detectable by the fSCA mapping techniques employed in this paper. I was hoping that these issues may have been considered in the discussion (e.g. sec. 4.6).

We thank the reviewer for bringing this issue up. Given the published MAE of 8 cm in snow depth (Painter et al. 2016), we find that snow cover cannot be reliably classified by ASO below this value. This 8 cm MAE is higher than values more recently reported by ASO Inc, starting in the water year 2021, which are 2-3 cm, validated with in situ measurements. However, the 8 cm value is lower than the 10 cm MAE reported by Currier et al. (2019) who compare gridded ASO snow depth with gridded snow depth maps from a terrestrial laser scanner. To our knowledge, these validations have taken place in relatively flat areas. We agree with the reviewer that validation in steeper terrain is needed, but is beyond the scope of this study.

For this work, we relied on two prior peer reviewed papers where we used the same ASO validation methodology and validated alongside co-located Worldview 2/3 datasets. Because we had already compared subsets of this ASO dataset to WV2/3 data, we did not duplicate that analysis in this paper. Those papers are citied in the submission:

Bair, E. H., Rittger, K., Davis, R. E., Painter, T. H., and Dozier, J.: Validating reconstruction of snow water equivalent in California's Sierra Nevada using measurements from the NASA Airborne Snow Observatory, 850 Water Resour Res, 52, 8437-8460, 10.1002/2016wr018704, 2016.

Bair, E. H., Stillinger, T., and Dozier, J.: Snow Property Inversion From Remote Sensing (SPIReS): A Generalized Multispectral Unmixing Approach With Examples From MODIS and Landsat 8 OLI, Ieee T Geosci Remote, 59, 7270-7284, 10.1109/Tgrs.2020.3040328, 2021a.

Our methodology is based on prior validation, from the above studies, with the ASO datasets. We acknowledge that the validation data are imperfect (original submission line 321-328, line 687-697, Line 749-752). For steep, rugged, terrain, past validation analysis of the difference

between 0.5m binary snow maps and ASO yielded an overall accuracy of 0.9998, F-stat of 0.991, recall of 0.992 and precision of 0.990 (Bair et al 2016, section 4.4.2 and table 4).

Only 1.9% of ASO snow depths has values less than 8 cm. Further, coarsening our validation datasets to 120 m and 2 km reduces the influence of spurious misclassifications of thin snow as false negatives. Using a different threshold would not significantly change the results of our analysis.

Technical/editorial comments

Line 27: "…spectral mixture…" should be spectral unmixing?

Changed.

Line 42: "…billions of people." This reads a little imprecisely/colloquially, and repetitive with respect to Line 2.

Changed - the second use of billions of people was deleted.

Line 61: "At a 463 m spatial resolution…" some further explanation/reference here would be useful, as many readers would expect to see 500 m here, as widely documented, and most users are typically interacting with data resampled to a 500 m grid. It is also worth noting that it is possible to map sow from MODIS at finer (e.g., 250 m) resolution.

While two bands are available at 250m, they are not in portions of the electromagnetic spectrum that enable a NDSI calculation, nor are they used in the standard products. While snow can be identified in some situation with these bands, we focused on the standard products and spectral unmixing methods for this paper as these are the best automated globally applicable algorithms that readers are most likely to come across.

Figure 1 (caption): "Pixels are 463 m." This statement is a little ambiguous, perhaps make explicit reference to the spatial resolution?

Spatial resolution has been added to the sentence.

Line 110: "MODiMlab" should be MODImLab?

Thanks for catching this error, changed.

Figure 3: I think this figure would benefit from some additional context, especially for readers outside of North America. Perhaps major basin outlines and/or elevation contours (at an appropriate interval for scale), for example, could be added?

Thank you for the recommendation. Readers outside of North America will be unfamiliar with the specific basins used in the study. The goal of the figure is not to highlight specific basins but instead to place the study locations in the geographic context of the western United States and highlight the diversity of snow classes within the study area. We have decided to leave Figure 3 unchanged.

Table 2: Reference Hall et al., (2019) missing from reference list? Presumably the document referred to is: MODIS Snow Products Collection 6.1 User Guide? Which would be Riggs et al. (2019). In any case, only refers to the spatial resolution being 500 m.

Please see our response above RE 463 m vs 500 m. References have been updated in the manuscript.

Line 207: "…(see Fig. 4)." Should be figure 2?

Thanks for catching this error, changed.

Line 304: what changes beyond the basin boundaries to make data unreliable?

The second paragraph in section 4.2.1 highlights the main issue- that the value of 0 in the 3m snow depth dataset represents both a NaN value and a 0 cm snow depth value in many files. Extensive manual examination of the ASO 3m snow depths alongside the final ASO product, the 50m SWE product, confirmed that this data representation issue is not present in the 50m SWE datasets and that by using the method described in line 306 – using the 50m SWE dataset to select valid 3m snow depth (and snow free) areas we were able to use the NaN values in the 50m SWE dataset to accurately map NaNs in the 3m snow depth dataset. Because the 50m SWE datasets are constrained to the basin boundaries, we did not have a reliable way to determine the difference between NaN and 0 snow depth outside of the basin boundaries.

Line 308 – 314: I'm not sure that the case is well made here for the 8 cm threshold – see previous comment.

Please see our response to the previous response on this topic as well. We have added additional text to clarify the reasoning behind the choice and past peer reviewed validation efforts using this dataset and this threshold. Some additional information on the characteristics of the data that is excluded is presented as well.

Line 314 – 315: How was the resolution coarsened? The choice of technique will impact results, so please be specific.

The following text has been added to these lines to clarify the coarsening procedure: *"The 3m binary snow cover maps were coarsened via gaussian pyramid reduction to reach the significantly coarser validation resolutions and then bilinear interpolation was used to reprojection the coarsened data to each snow products native projection."*

Line 324 – 328: Mirroring my earlier comment, given the importance of ASO data to the rest of your analysis, I think you could go further in evaluating its quality and considering limitations here.

The validation of ASO data is beyond the scope of this paper and has already been performed as noted in earlier comments. Limitations are discussed at the end of section 4.2.1. See Painter 2016, Currier 2019, 2021-onwards ASO reports for evaluations.

Line 384: It could be useful for readers to indicate the range of typical values you see here for F statistic?

The range for F values has been added to the paper.

Line 469: "All snow cover fractions…" is all the right word here? Implies all possible outcomes occurred/observed, perhaps indicate the range of observed fSCA?

Yes this is correct, all possible snow cover fractions were observed in all snow cover products.

Line 484: "…constricted…" would constrained or restricted be better here?

Constricted has been replaced with "constrained".

Figure 4: x axes should be labelled on plots.

Labels have been added to the x-axes in the figures. X-axes are the same for all the graphs in each column, same for y axis for each graph in each row. This had been clarified in the caption for figure 4 and figure 5 which both use the same approach.

Line 545: Collection of ASO data suggests good viewing conditions, but is there no possibility of cloud impacts – seems like a question worth asking if maritime regions see highest RMSE?

Validation was only performed on cloud free days for Landsat scenes, and ASO is flown on clear days and underneath most clouds. Any day where clouds were present was removed from the validation dataset as described in line 168.

Line 548: "…slightly edging out…" perhaps outperforming would be better phrasing here?

Changed.

Line 732 – 734: Is it realistic that there might be no issues with the reference data contributing here?

Yes, you are correct that there could be issues with the reference data. ASO may not map all rock outcrops correctly in alpine regions and may instead consider those locations to be fully snow covered. So, ASO might be overestimating snow cover at high fSCA values, contributing to the negative bias we measured. But, based on our analysis of the spectral unmixing solutions for these high snow cover alpine pixels and past comparison of ASO to Worldview 2/3 snow maps in Bair et al 2016, we are confident that the reasons posited at the beginning of section 6.4 are the dominant factors in the negative biases found for these high snow cover pixels. Theses are A) the combination of positive bias being impossible to achieve (cannot have more than 100% snow cover) and B) the "one-to-many" issue

with spectral unmixing where solutions with fSCA close to 100% generates plausible spectral unmixing solutions (e.g. fSCA of >95% but ~=100%). These are reasonable issues that transcend specific validation methodologies and are applicable to any spectral unmixing approaches for any land surfaces. We think it is important for the reader to be aware of and focus on these points over ASO data details.

References

Painter, T.H., Berisford, D.F., Boardman, J.W., Bormann, K.J., Deems, J.S., Gehrke, F., Hedrick, A., Joyce, M., Laidlaw, R., Marks, D. and Mattmann, C., 2016. The Airborne Snow Observatory: Fusion of scanning lidar, imaging spectrometer, and physically-based modeling for mapping snow water equivalent and snow albedo. Remote Sensing of Environment, 184, pp.139-152.

Riggs, G.A., Hall, D.K. and Román, M.O., 2015. MODIS snow products collection 6 user guide. National Snow and Ice Data Center: Boulder, CO, USA, 66.

---

## Author Response (AR1)

Updates requested by the editor have been included in the track changes version in addition to the updates in our responses to the reviewers.

Additionally, the line style for figure 4 was updated to be more colorblind friendly.

SPIReS and SCAG datasets will have DOIs created for final submission. Line 827 and 831 in the revision file will be updated with final DOIs once they are formalized.